# Vision CNNs trained to estimate spatial latents learned similar ventral-stream-aligned representations

**Yudi Xie***  **Weichen Huang**  **Esther Alter**  **Jeremy Schwartz**
**Joshua B. Tenenbaum**  **James J. DiCarlo**

Department of Brain and Cognitive Sciences, MIT

## Abstract

Studies of the functional role of the primate ventral visual stream have traditionally focused on object categorization, often ignoring – despite much prior evidence – its role in estimating "spatial" latents such as object position and pose. Most leading ventral stream models are derived by optimizing networks for object categorization, which seems to imply that the ventral stream is also derived under such an objective. Here, we explore an alternative hypothesis: Might the ventral stream be optimized for estimating spatial latents? And a closely related question: How different – if at all – are representations learned from spatial latent estimation compared to categorization? To ask these questions, we leveraged synthetic image datasets generated by a 3D graphic engine and trained convolutional neural networks (CNNs) to estimate different combinations of spatial and category latents. We found that models trained to estimate just a few spatial latents achieve neural alignment scores comparable to those trained on hundreds of categories, and the spatial latent performance of models strongly correlates with their neural alignment. Spatial latent and category-trained models have very similar – but not identical – internal representations, especially in their early and middle layers. We provide evidence that this convergence is partly driven by non-target latent variability in the training data, which facilitates the implicit learning of representations of those non-target latents. Taken together, these results suggest that many training objectives, such as spatial latents, can lead to similar models aligned neurally with the ventral stream. Thus, one should not assume that the ventral stream is optimized for object categorization only. As a field, we need to continue to sharpen our measures of comparing models to brains to better understand the functional roles of the ventral stream.

## 1 Introduction

David Marr famously introduced vision as "knowing what is where by looking" (Marr, 2010). The core functions of vision involve recognizing "what" an object is (object recognition) and determining "where" it is by estimating the spatial latent variables of the object, such as its position and pose. Humans and primates excel at object recognition (DiCarlo et al., 2012) and are clearly capable of estimating spatial properties like object position or pose from visual input (Hong et al., 2016). Traditionally, it has been believed that the "what" and "where" functions of vision are dissociated (Schneider, 1969) and supported by two different cortical pathways, with the ventral stream primarily associated with the "what" function of object recognition (Mishkin et al., 1983). However, the exact nature of the "what" vs. "where" dissociation is debated (Goodale & Milner, 1992). Many previous studies found that the ventral stream encodes not only object identity ("what") but also rich object spatial information ("where") (Connor & Knierim, 2017), such as object position and pose (Hong et al., 2016; Hung et al., 2005). This indicates that interpreting the ventral stream's function as purely object recognition may be an oversimplification. The exact functional role of the primate ventral stream remains to be fully understood.

---

*Correspondence: yu_xie@mit.edu
Code and datasets: https://github.com/YudiXie/multitask-vision

Despite the known role of the ventral stream in representing object spatial information, leading CNN models of this pathway are mostly optimized for object categorization tasks, leaving spatial objectives underexplored. In this study, we ask: Might the ventral stream be optimized for spatial tasks? We examine how training CNNs to estimate spatial latent variables – such as object position and pose – affects their alignment with neural responses in the primate ventral visual stream. This area remains less explored largely due to the scarcity of large-scale datasets with comprehensive spatial labels beyond categories. To overcome this, we use a 3D graphic engine to generate large-scale synthetic image sets that contain detailed information on various spatial latent variables (Gan et al., 2020). In most analyses, we trained CNNs to predict different spatial latent variables while keeping their architecture and training data constant, ensuring that any differences in their learned representations are attributable to the training objective. After training, we evaluated the learned representations and their alignment with primate ventral stream neural data using various methods, including the Brain-Score open science platform (Schrimpf et al., 2018).

We found that the neural alignment of models trained on synthetic images is close to those trained on natural images, although their entire training experience has been purely on synthetic images. To our surprise, we found training models on spatial latent variables is sufficient to drive up neural alignments with the primate ventral stream, as models trained to estimate just a handful of spatial latent variables have neural alignment scores comparable to models trained on a much larger number of categories. One might speculate that their similar alignment could be due to either (1) they indeed learn similar representations that support similar neural alignment, (2) or their learned representations are different, but the alignment metric is not sensitive enough to tell the differences between these models. Thus, we asked the question: How different – if at all – are representations learned from estimating spatial latents or categories compared to each other? We used centered kernel alignment (CKA) (Kornblith et al., 2019) to analyze the similarity between models trained on different latents. We found that, despite being trained on dramatically different objectives, these models learned very similar – but not identical – internal representations, especially in their early and middle layers.

This observation raises the question: Why do these models learn similar representations despite being trained to estimate very different targets? One hypothesis is that when models are trained to learn some target latents in a dataset that has variations in other non-target latents, they inadvertently learn to represent those non-target latents as well, rather than developing representations that are invariant to these non-target latents. This may happen even though the models are not explicitly trained to estimate those non-target latents. We provided evidence supporting this hypothesis by analyzing the learned representations while controlling the non-target latent variability in the training data. This partially explains why models learn similar representations, as learning a subset of the latents will inadvertently learn a converged representation that supports the inference of all other latents.

In summary, our findings are as follows:

- CNNs trained on purely synthetic image datasets learned representations with neural alignment scores close to those trained on natural images.
- CNNs trained to estimate only a handful of spatial latents achieve neural alignment comparable to those trained on hundreds of categories. Models' neural alignment correlates strongly with their spatial performance.
- The learned representations of CNNs trained to estimate different spatial and category latents are similar (but not identical), especially in their early and middle layers.
- Non-target latent variability helps models learn better representations of non-target latents rather than become invariant to them, providing a partial explanation for why models develop similar representations.

## 2 METHODS

### 2.1 IMAGE DATASETS GENERATED BY A 3D GRAPHIC ENGINE

Investigations on neural network models trained on spatial latent variables are often hindered by the lack of large-scale image datasets with comprehensive and accurate labels of the spatial la-

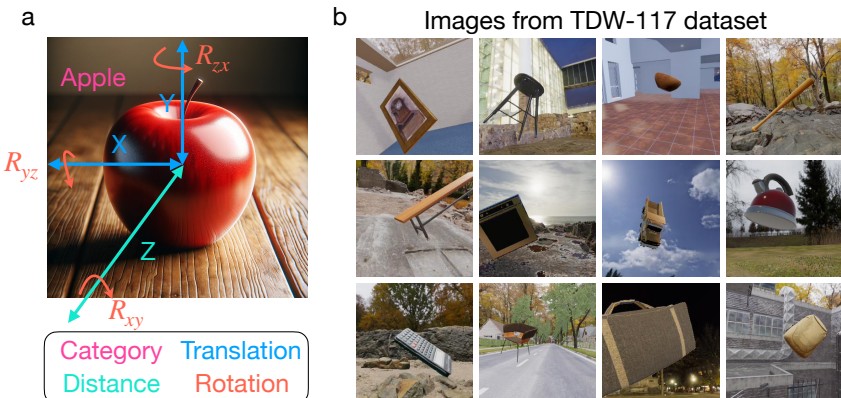

Figure 1: **Spatial latent variables and our training dataset.** **(a)** An illustration of the set of spatial latents that are available in the dataset for training models. In addition to the object category (Apple), the set of spatial latent variables we record are the following: translation (X, Y), distance (Z), and rotation ($R^{xy}$, $R^{yz}$, $R^{zx}$). (This image is for illustration only, not in the synthetic dataset.) **(b)** Example images in our dataset (TDW-117) for training CNNs. Each image contains one object with varying positions and poses against a random background.

tent variables. Most previous studies on the neural alignment of vision models have focused on training models with categorization or various self-supervised training objectives. The effect of training on spatial latent variables remains less understood. To overcome this problem, we generated several synthetic image datasets using ThreeDWorld (TDW) (Gan et al., 2020), a Unity-based 3D graphic engine (Figure 1 b). We generated image datasets that contain up to 100 million images from 117 object categories made up of 548 specific object 3D models (on average, about 5 object models per category). Each image in the dataset contains one object rendered with a random position and pose against a random background. We record the ground truth information of a set of latent variables for each image (Figure 1 a); these include the horizontal position $x$, vertical position $y$, distance to the center point of the object $z$, and the three Euler angles quantifying the object's rotation related to a pre-defined orientation of each object model $\{R^{xy}, R^{yz}, R^{zx}\}$. All these latent variables are recorded in the spatial reference frame tied to the camera. We also record each image's broader object category $C^{cat}$ and the specific object identity $C^{id}$ in addition to spatial latent image variables. Together, we call them the latent set for each image in the dataset $\{X, Y, Z, R^{xy}, R^{yz}, R^{zx}, C^{cat}, C^{id}\}$, which contains both spatial and category latents. For more details about the image generation process, see Appendix A.

## 2.2 NEURAL NETWORK TRAINING OBJECTIVES

We used the ResNet architecture family for visual estimation (He et al., 2016). Specifically, we focused on ResNet-50 and ResNet-18. We train models to estimate different subsets of the spatial or category latents conditioned on the image using supervised learning. For discrete targets like $\{C^{cat}, C^{id}\}$ in the latent set, we used the cross-entropy loss. For continuous variables in the latent set, like $\{X, Y, Z, R^{xy}, R^{yz}, R^{zx}\}$, we used the square loss. Distance, translation, rotation, object category, object identity models are trained to estimate $\{Z\}$, $\{X, Y\}$, $\{R^{xy}, R^{yz}, R^{zx}\}$, $\{C^{cat}\}$, $\{C^{id}\}$ respectively. When these objective are combined, such as "Distance + Translation", we take the arithmetic mean of the loss function of each task in the combination as the total loss. We used mini-batch stochastic gradient decent with Adam optimizer (Kingma, 2014) for training the neural networks. Models are trained until they reached a plateau in test performance in a held out test set. For more details on the architecture, training, and evaluation of the neural network models, see Appendix B.

## 2.3 ANALYSIS OF THE LEARNED REPRESENTATIONS

**Neural alignment metrics**. After training each model to estimate spatial latents or categories, we evaluate how well models align with the primate ventral stream neural data at four stages of

the ventral stream hierarchy (V1, V2, V4, IT). We used the Brain-Score open-science platform to calculate models' neural-alignment scores with each of the four brain regions (Schrimpf et al., 2018). The scores are numbers in the range of [0, 1], with 1 meaning the closest alignment between the model and data in the benchmarks. These scores measure how well the internal representations of models can predict neural responses in the corresponding brain regions on held-out images. For analyses in Figure 2 and Figure 3, we plotted the model's scores averaged over four different neural regions (V1, V2, V4, IT) as the overall alignment. For more details about the neural alignment measures, see Appendix C.

**Similarity of neural representations**. In addition to measuring the models' alignment with the ventral visual stream, we investigated a closely related question: How different are representations learned from different spatial latent estimation tasks compared to each other and compared to models trained on categorization? To quantify the similarity between the representations of different models, we used centered kernel alignment (CKA) (Kornblith et al., 2019), which has gained popularity due to its thoughtful consideration of the invariant properties of similarity metric and ability to reveal similarity between models trained with different random initialization. To compute CKA, we extract models' activation at different layers spanning from early to late layers in response to 2000 held-out images in the TDW-117 image dataset. For more details, see Appendix D

## 3 RESULTS

### 3.1 LEARNING A SMALL NUMBER OF SPATIAL LATENTS PRODUCES NEURAL-ALIGNED CNN MODELS OF THE VENTRAL STREAM

We optimized the CNNs to estimate specific subsets or the full set of latent image variables for which we have ground-truth supervised targets from the graphic engine. These variables include spatial latents such as distance, translation, rotation, category information such as object category and identity, and combinations thereof (see Figure 2 b). First, we found that models optimized purely on the synthetic image dataset (TDW-117) achieved neural alignment scores very close to the same CNN architecture trained on the natural image dataset, ImageNet. For example, models optimized to classify object categories in the synthetic dataset achieved more than 95% of the neural alignment score of the ImageNet-trained model (Object category classification: 0.412 compared with ImageNet-1K: 0.430). This suggests that synthetic image sets are effective in driving model-brain alignment, even though they do not capture all the nuances of natural images. Similar results were found in another model architecture, ResNet-18 (Table C.2).

We sorted these models by the number of latent variables they were optimized to predict, which corresponds to the number of output units receiving supervision (see Figure 2). For instance, a model trained to estimate object distance has only one output unit, while a model trained to estimate object category has 117 output units. We then investigated the neural alignment scores of these models. Intuitively, one might predict the "scaling hypothesis," which suggests that the more latent variables a model is trained to predict (and thus the more supervision it receives during training), the more neurally aligned it would become. This is true for models trained to do object categorization with datasets that have different numbers of categories (TDW-N), as the neural alignment scores of these models scale with the number of categories in the dataset. However, to our surprise, we found that models trained to estimate spatial latents do not adhere to the "scaling hypothesis." Many CNNs trained to estimate a very small number of latent variables achieved surprisingly high neural alignment scores, comparable to models trained on over a hundred object categories or more than five hundred object identities (Figure 2 a). For example, CNNs trained to estimate the distance of an object to the camera, which receive supervision from only one output unit, achieved a neural alignment score of 0.399 – 97% of the score attained by CNNs supervised on all classification tasks and the full set of latent variables (0.411). Similar results were found in another model architecture (Table C.2).

We also investigated the behavioral alignment of spatial latent trained models. We found their behavioral alignment scores are not as good as category-trained models (Figure C.1 f, Figure C.3 f). The behavioral benchmark we used focuses on the consistency between models' behavioral readout and primate behavior in categorization tasks. Thus, it may be biased toward category-trained models. Further studies are needed to quantify whether spatial latent trained models are more behavioral aligned than category-trained models in spatial latent estimation tasks.

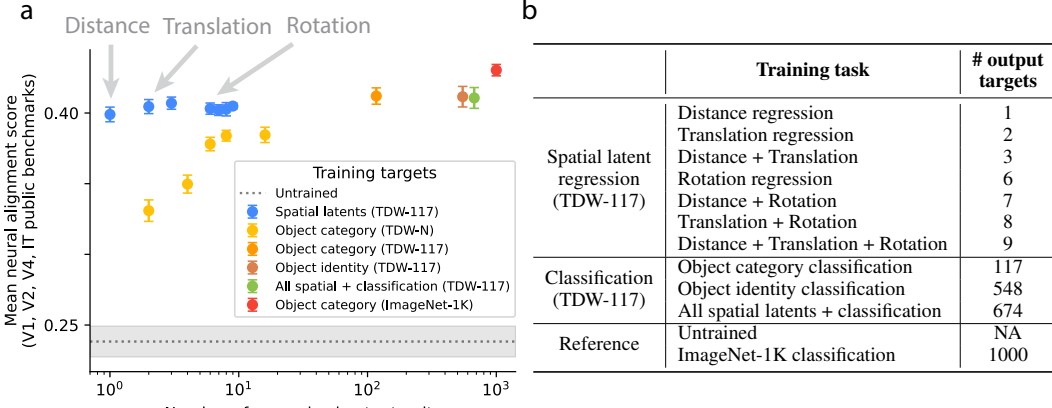

Figure 2: **Learning a small number of spatial latents produces ventral-stream-aligned CNN models.** **(a)** The neural alignment of models (ResNet-50) trained on different objectives (x-axis, number of output units, see panel b). Learning a few spatial latent variables produced models that have neural alignment scores comparable to models trained on hundreds of categories. Error bars or shaded regions show the SD across multiple random seeds (N=5). All spatial + classification = all spatial and all classification tasks combined. TDW-117 is our TDW dataset with 117 object categories; TDW-N means datasets with N = 2,4,6,8,16 categories. For a breakdown of the individual region alignment scores, see Figure C.1. **(b)** The training tasks we investigated and their corresponding number of output units that receive supervision during training.

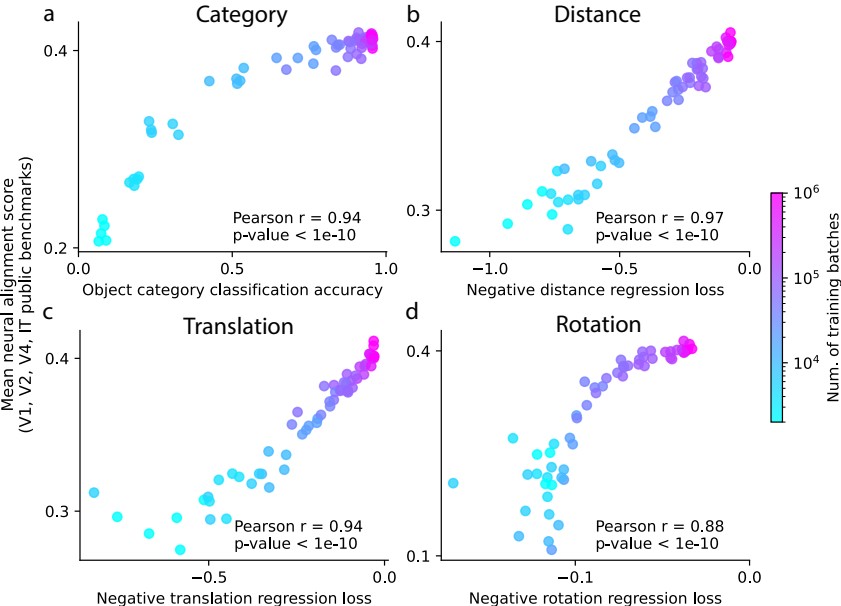

Figure 3: **The neural alignment of models correlates strongly with their spatial task performance.** **(a)** The neural alignment scores correlate with models' categorization performance for models trained on object categories. This figure shows results from multiple random initializations. Each dot shows a ResNet-50 model colored by the number of training batches. **(b-d)** Models' neural alignment scores correlate with their spatial latent estimation performance when they are trained to estimate those latents respectively. The figures show models trained on **(b)** distance regression, **(c)** translation regression, **(d)** rotation regression (1 outlier out of 60 data points where the loss is larger than 0.2 is excluded). For a breakdown of the averaged score into individual scores, see Figure C.7 and Table C.3.

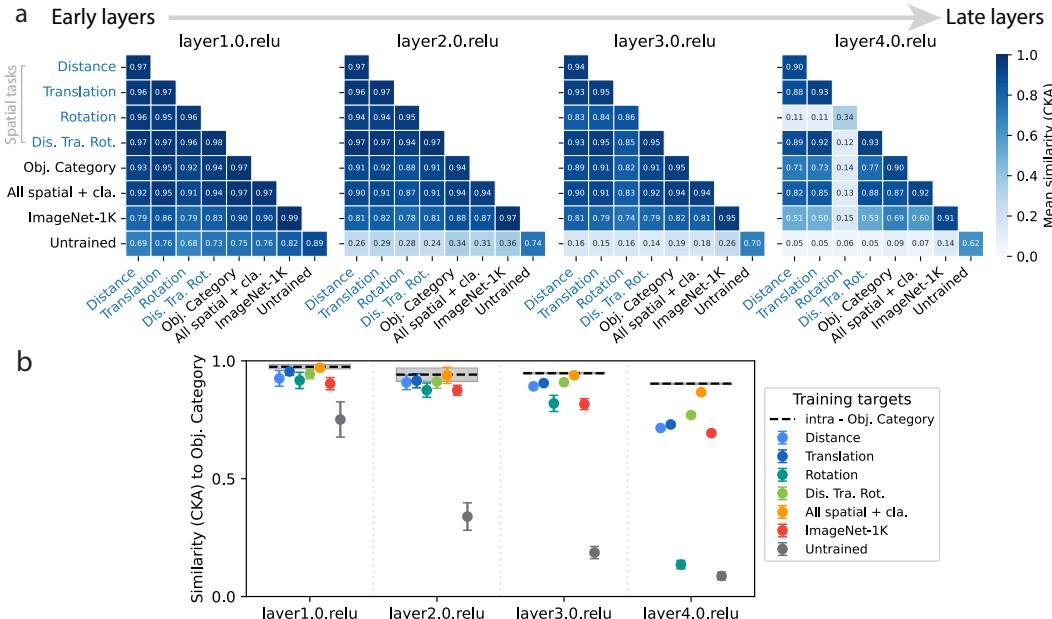

Figure 4: **Models trained to estimate different latents learned similar representations. (a)** Pairwise similarity (CKA) between models trained on different targets at 4 different layers. Until the last few layers (e.g., layer4.0.relu), the representations of models trained on different spatial and category latents remain highly similar. Off-diagonal entries are the averaged pair-wise similarity between models in the two groups. Diagonal entries are averaged similarity between different randomly initialized models in the same group. (All spatial + cla. means models trained on all spatial and classification tasks) **(b)** The similarity between category classification models and models trained on other targets. Models trained on spatial latents remain similar to category models until the last few layers. Layer names as in Table B.1. intra - Obj. Category shows the averaged distance between categorization models trained with different random initializations.

Previous studies about the functional roles of the primate ventral stream have shown that a CNN's object categorization performance strongly predicts its ventral-stream neural alignment (Yamins et al., 2014; Schrimpf et al., 2018). This result is often interpreted as implying that evolution and/or postnatal development derived the primate ventral stream under the objective of object recognition. Our experiments showed similar results in models trained for categorization (Figure 3 a). However, we found that models' performance on spatial latent estimation strongly correlates with their neural alignment as well when they are trained to estimate those latents respectively (Figure 3 b-d). These results support the interpretation that the primate ventral stream function is also to estimate these spatial latent variables. Thus, we should not simply assume that the ventral stream is optimized for object categorization only. Instead, these results support that the ventral stream representations should be considered a set of general-purpose visual representations that can support both object categorization and spatial estimation tasks.

Finally, we investigated how the neural alignment of these latent trained CNNs scales with the size of the dataset used for training. We found that the neural alignment score of CNNs trained on TDW images increases logarithmically with dataset size in a low data regime (Figure C.2). However, this trend reaches a plateau at around 1-10 million images; further increasing the training dataset size does not increase their neural alignment further.

## 3.2 MODELS TRAINED TO ESTIMATE DIFFERENT LATENTS LEARNED SIMILAR REPRESENTATIONS

Our results suggest that models trained on different spatial latents achieve similar alignment scores with the primate ventral stream. Two possibilities could explain this result: First, the models may learn dissimilar representations, either because the neural alignment metric fails to distinguish them,

or because they explain non-overlapping variance in the neural data, leading to similar overall scores. Second, the models might learn similar representations, naturally resulting in similar alignment scores. To further differentiate these two possibilities, it is important to investigate how different – if at all – are representations learned from spatial and category latent tasks compared to each other. So, we examined the similarity of the learned representations across models trained on different tasks while holding the training data and model architecture constant.

We analyzed the representation similarity using centered kernel alignment (CKA) (Kornblith et al., 2019). We found that models trained to estimate different subsets of the latent variables learned very similar representations in early (layer1.0) to mid-layers (layer2.0, layer3.0) and started to diverge in late layers (layer4.0). Models trained to estimate rotation showed the most significant divergence from other models (Figure 4 a, Figure D.3). Models trained to estimate spatial latents remain similar to categorization-trained models until the last few layers (Figure 4 b, Figure D.1). In early to mid-layers, models trained to estimate different latents often are not more different than models trained to estimate the same latents but initialized with different random seeds (Figure D.2).

These results suggest that models trained on different targets can learn surprisingly similar representations, especially in the early and intermediate layers. Meanwhile, our findings in Section 3.1 show that models trained with combined objectives (e.g., Distance + Translation) do not achieve much higher neural alignment scores than those trained on individual latents (Figure 2, Table C.1). If models trained on different latents learned largely distinct representations, explaining non-overlapping variance in neural data, one would expect that combining these latent targets would produce a more aligned model. However, this was not observed. Taken together, our results suggest that the similar neural alignment scores we observed may mainly be attributed to the similar representations these models learned. However, it is worth noting that, despite these similarities, differences still exist between models trained on different targets, especially in late layers. It is likely that the current brain-model comparison measures are not sensitive to many subtle differences that exist among these models.

### 3.3 Non-target latent variability helps learn better representations of the joint latents

We further asked a closely related question: Why did models trained to estimate different spatial and category latents learn similar representations as assessed by CKA? Although these models are trained to estimate different *target latents*, they are all exposed to the same dataset, which contains variability across all latent dimensions, most of which are not the training target. For example, distance is the target latent for a model trained to estimate distance. Translation, rotation, and latents other than distance are *non-target latents*. The variability in the dataset's non-target latents may play an important role in shaping the learned representations. One hypothesis is that when models are trained to estimate some target latents, the non-target latent variability may help the models inadvertently learn to represent those non-target latent variables as well. Consequently, when models are trained on a dataset with variability across all latent factors, they may converge to a representation that supports the inference of the entire set of latents, thereby explaining the similarity in representations across models trained on different tasks.

To test this hypothesis, it is crucial to determine whether non-target latent variability indeed facilitates the learning of these non-target latents. Using a synthetic dataset, we performed causal manipulations to investigate the impact of non-target latent variability on the learned representations. Our experimental design, illustrated in Figure 5 a, involves measuring how well a model has learned to represent a latent variable by assessing the linear decoding performance of that latent at different layers in a neural network. For instance, even if a model is trained to estimate a specific target latent (e.g., translation), it may still allow for decoding other non-target latents (e.g., category) at various layers. We established a baseline by training models with reduced non-target latent variability (e.g., a dataset with only one category) and compared its non-target latents decode performance to models trained with full non-target latent variability (e.g., a dataset with multiple categories). We consider two possible outcomes: (H1) the added non-target latent variability facilitates the learning of that latent, leading to improved decoding performance even though the model is not explicitly trained to estimate it; or (H2) the model's representation becomes more invariant to the non-target latent, as it may be irrelevant to the task of estimating the target latent.

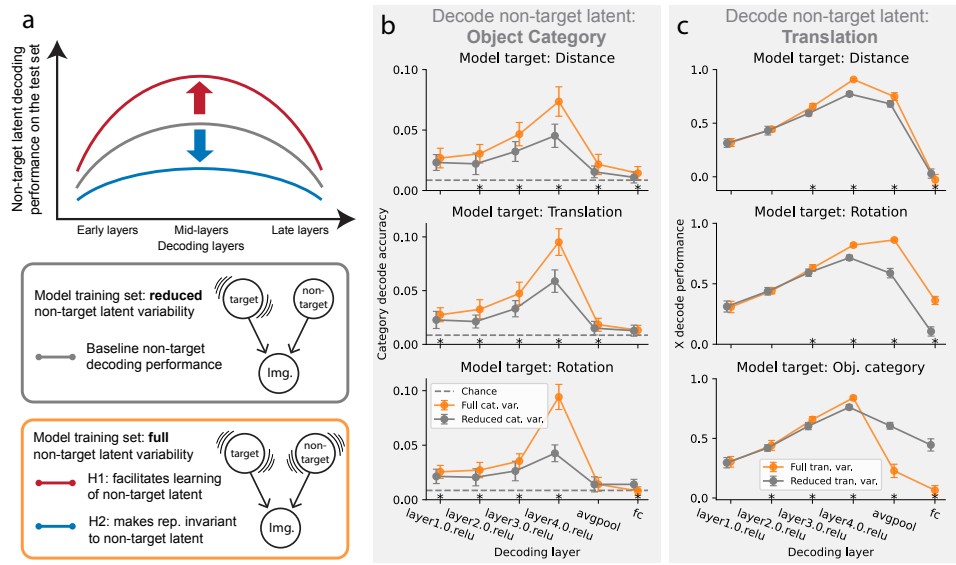

Figure 5: **Non-target latent variability helps learn better representations of the joint latents in intermediate layers. (a)** We compared the decoding performance of non-target latents at different layers of the models. Using a model trained to estimate some target latents on a dataset that has reduced non-target latent variability as a baseline, we can expect two different outcomes. First (**H1**), if a model trained with full non-target latent variability decodes the non-target latents better, that indicates that the non-target latent variability facilitates the learning of that non-target latent, although the models are not trained to estimate them. Second (**H2**), it is also possible that additional non-target latent variability makes models become invariant to that non-target latent, thus worse decoding performance for that non-target latent. Our experiments suggested that models learned better representations of the non-target latent with additional non-target latent variability, supporting H1. **(b)** Decoding performance for the non-target latent – category – when the models are trained to estimate target latents – distance (top), translation (middle), rotation (bottom). Non-target latent variability helped models learn better representations of these latents in the intermediate layers. (cat. var. – category variability). **(c)** Similar results were seen when the non-target latent is translation, and the target latent is distance (top), rotation (middle), and category (bottom). When the target latent is category, models learn better representations of translation with additional translation variability, although they became more translation invariant in the last two layers. (tran. var. – translation variability). The results for another translation latent Y is similar (Figure E.2). (Error bars show the SD across 5 cross-validation runs $\times$ 6 randomly initialized models. "*" indicates a significant difference between the two groups, Mann-Whitney U test, p value $< 0.05$). Layer names as in Table B.1.

In our analysis, we compared models trained on spatial tasks with a dataset of reduced category variation against those trained with full category variation (Figure 5 b). Despite not being trained to estimate the non-target latent (category), the added variability led to improved representations of that non-target latent in the intermediate layers (layers 2, 3, and 4), supporting H1. (For a comparison of the category decoding performance of spatial latent-trained models to that of category-trained and untrained models, see Figure E.1.) Similar results were observed when we used translation as the non-target latent and manipulated its variability (Figure 5 c). An exception is in models trained to estimate object category (Figure 5 c, bottom), where increased translation variability led to more invariant representations in the last two layers (avgpool and fc), supporting H2. However, even in these cases, intermediate layers still showed improved representations of translation, supporting H1. Moreover, we found that models trained with full non-target latent variability often learned representations that are more similar to models trained directly to estimate those non-targets (Figure D.4). More non-target latent variability may help models learn more similar representations within each group of models trained on the same target but initialized differently (Figure D.5, Figure D.6). In addition, we found that models trained with full non-target latent variability achieved better neural alignment scores when the non-target latent was category (Figure E.3), though this effect was less pronounced when the non-target latent was translation.

In summary, most of our observations are consistent with the hypothesis that the variability of non-target latents helps develop better intermediate representations of them, though future work is needed to understand both the size and mechanism behind this effect. The variability across all latents in the dataset likely aids the models in learning representations that facilitate the inference of joint latents. This partly explains the similarity in intermediate representations across models trained on very different targets, as observed in our previous results. However, multiple factors likely contribute to this similarity. While we have identified non-target latent variability as one such factor, others – such as rendering quality and the statistical properties of image stimuli – may also play a significant role in shaping model representations and need further investigation.

## 4 RELATED WORKS

**The impact of task choice on neural alignment of vision models**. Previous studies found CNNs trained on object categorization tasks strongly predict neural responses in the primate ventral stream (Yamins et al., 2014; Khaligh-Razavi & Kriegeskorte, 2014). Subsequent studies have predominantly focused on studying how different model architectures impact neural alignment (Schrimpf et al., 2018; Kubilius et al., 2019; Kar et al., 2019; Storrs et al., 2021) while keeping the training and evaluation task fixed at object classification. More recent studies have begun to explore the effects of alternative training objectives on neural alignment. Zhuang et al. (2021); Konkle & Alvarez (2022); Yerxa et al. (2023) investigated models trained with self-supervised learning objectives and showed that they are comparable to models trained on supervised classifications. Conwell et al. (2022) examined various tasks from the Taskonomy dataset (Zamir et al., 2018), such as semantic segmentation and depth-map estimation. Our study continues in this trend of investigation but focuses on estimating object-centric 3D spatial latents. These objectives can be framed as intuitive tasks humans can naturally do; how these objectives impact models' neural alignment is unknown. In addition, many self-supervised learning objectives often push models to learn representations that minimize the differences between different augmentations of the images, such as random cropping and resizing, which may destroy the object-centric spatial latent information.

**Representation learning from synthetic datasets**. Synthetic datasets have become a valuable tool in computer vision for addressing limitations associated with real-world data. Synthetic data has been effectively used to train deep neural network models and has shown to benefit performance on semantic segmentation, object detection, pose estimation, depth, and optical flow estimation (Richter et al., 2016; Ros et al., 2016; Mayer et al., 2016; McCormac et al., 2017; Varol et al., 2017). Models trained with synthetic images from generative models learned visual representations that rival the performance of those trained on large-scale natural image dataset (Azizi et al., 2023; Tian et al., 2024b;a). However, how well representations learned from synthetic datasets align with neural processes in the brain remains less understood. Our study showed that models trained exclusively on synthetic images can achieve neural alignment scores close to those trained on natural images.

**Similarity of neural representation and convergence**. Many previous studies developed various methods to analyze the similarity of representations learned in neural networks (Raghu et al., 2017; Morcos et al., 2018; Kornblith et al., 2019; Dravid et al., 2023; Lenc & Vedaldi, 2015; Bansal et al., 2021). Significant similarity can be found between representations of networks trained with different random initialization (Li et al., 2015; Kornblith et al., 2019; Bansal et al., 2021), different network architectures trained on the same task (Laakso & Cottrell, 2000; Raghu et al., 2017; Kornblith et al., 2019), networks trained with different dataset (Lenc & Vedaldi, 2015; Kornblith et al., 2019; Bansal et al., 2021), and between models trained with supervised and self-supervised objectives (Bansal et al., 2021), although these results depend heavily on the specific definition of similarity. CKA has gained popularity partly due to its thoughtful consideration of the invariant properties of the similarity metric and its effectiveness at revealing the similarity between representations trained with different random initialization (Kornblith et al., 2019). We used CKA to measure similarity and have shown that models trained to estimate very different spatial latents can also learn similar representations. Earlier layers are more similar than later ones, consistent with previous findings in classification models trained with different random initialization (Kornblith et al., 2019). Meanwhile, there is a trend that higher-performing, wider, and larger networks learned more aligned representations (Morcos et al., 2018; Kornblith et al., 2019; Bansal et al., 2021), even representations of vision and language models could become more similar as performance increases (Huh et al., 2024). Huh et al. (2024) hypothesized that model representations may converge as they learn a better representation

of the reality of the world. Our experiments in Section 3.3 provide evidence that models may be learning the world even if the training task is only partially related to it.

## 5 DISCUSSIONS

In this work, we found that models trained to estimate a few spatial latents can achieve high neural alignment scores comparable to models trained on object categorization. Moreover, these models develop similar representations in the early to middle layers despite being trained on different latents. To understand why they converge, we explored the role of non-target latent variability in shaping the learned representations. Our analyses suggest that greater variability in non-target latents enhances the models' ability to represent these latents. This partially explains why models trained on different targets converge to similar representations, as they inadvertently learn to encode all other non-target latents. Our study on latent variability also connects to the broader literature in machine learning that explores the role of data diversity and augmentation in domain generalization (Arjovsky et al., 2019; Gulrajani & Lopez-Paz, 2020; Koh et al., 2021). Greater data diversity has been shown to improve out-of-distribution generalization (Tobin et al., 2017; Madan et al., 2022).

Our findings on the surprisingly strong neural alignment of spatial latent-trained models, along with the observed correlation between spatial latent performance and neural alignment, suggest that one should not assume the ventral stream is optimized for categorization only. Our findings in Section 3.2 indicate that computational models trained on either the "what" or "where" tasks may learn similar representations, which suggests that the separation of these functions may be an oversimplification. The inherent variability in the world may drive models to simultaneously represent both "what" and "where," learning one function may inevitably lead to learning the other. Meanwhile, we note that the representations learned from the different tasks we investigated are not identical. As a field, we need to continue to sharpen our measures of comparing models to brains, as it is likely that subtle differences exist among these models that our current comparison measures are not sensitive to.

One way to interpret our work is to view the trained CNNs as hypotheses for efficient visual inference models, which compress the complex computations involved in visual inference – typically modeled through generative processes – into an efficient feedforward pass (Yildirim et al., 2020; Dasgupta et al., 2020). We found that models trained on purely discriminative tasks to estimate some latents develop representations for other non-target latents, despite not being explicitly trained on them. This suggests that in order to learn certain objectives effectively, models may need to internalize the underlying generative structure of the world. For instance, object rotation is defined relative to a canonical view, which depends on object identity; thus, learning rotation also leads to learning the object category. However, we also found that models trained on translation tasks developed better representations of object categories with more category variability. The relationship between translation and object category is less intuitive, yet the models still learned both. This suggests a deeper connection between these functions, which was not previously known.

**Limitations and future works.** There are several limitations of the current work that invite future investigation. First, while the neural alignment of models trained on our synthetic dataset is comparable to, it still does not fully match, that of models trained on large-scale natural datasets such as ImageNet. Future work could explore whether improvements in rendering quality, greater scene and object diversity, or training on other natural datasets, where feasible, could enable models to match or even surpass those trained on ImageNet. Second, when evaluating our models on other non-public benchmarks in Brain-Score (Figure C.5 and Figure C.6), we found that models trained on spatial latent variables performed comparably to category-trained models in most V1, V2, and V4 benchmarks. However, among the five IT benchmarks, spatial latent-trained models significantly underperformed category-trained models in two benchmarks while outperforming them in one (Figure C.5). These three benchmarks are based on natural images that are out of the training distribution of our models. Further studies are needed to evaluate the extent to which our findings, primarily based on public benchmarks, generalize to a broader set of datasets and benchmarks. This remains an active area of research (Ren & Bashivan, 2024; Madan et al., 2024). Finally, our neural alignment evaluations rely heavily on regression-based metrics. Future research could incorporate alternative metrics, such as representational similarity analysis (Nili et al., 2014; Schütt et al., 2023), to assess whether our results hold under different analytical frameworks.

ACKNOWLEDGMENTS

We sincerely thank Kohitij Kar, Martin Schrimpf, Michael Lee, Chengxu Zhuang, Guy Gaziv, Robert Ajemian, and Lynn Sörensen for their valuable discussions and insightful suggestions on this project. This work was supported in part by the Semiconductor Research Corporation (SRC) and DARPA.

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

# APPENDICES

## A  DETAILS OF SYNTHETIC DATASETS GENERATED BY A GRAPHIC ENGINE

### A.1  GENERATION OF THE TDW-117 DATASET

We use ThreeDWorld Gan et al. (2020), a Unity-based 3D graphic engine, to generate the training dataset to train the latent estimation models from scratch. The image dataset we used to train the ResNet-18 and ResNet-50 models containing 1.3 million images from 548 specific object 3D models belonging to 117 object categories (used in Figure 2, Figure C.1, and Figure C.3). We called this TDW-117 dataset. To investigate how neural alignment scales with dataset size (Figure C.2), we generated a larger dataset that contains 100 million images. For all the datasets, we used 10 different 3D scene spaces including indoor and outdoor scenes. To generate each image, we randomly choose one camera position within the 3D scene space and randomly sample an object position and pose in a spherical space around the camera while making sure the object is not placed far below the camera. We then point the camera toward the object and randomly rotate the camera before taking an image. The above procedure ensures that there are variations in distance, object translation, and rotation related to the camera. We also randomly sample the skyboxes in the 3D engine to add additional variations in lighting and background. All objects are scaled to a uniform size so that their longer axis is the same.

For each image, we record the object center's position related to the camera's 3D spatial reference frame $(x, y, z)$, with $x$ and $y$ being the coordinate in the horizontal and vertical axis and $z$ is the coordinate in the axis pointing into the image. We used $x$ and $y$ for the translation regression task and $z$ for the distance regression task. Meanwhile, we also record the objects' 3D rotation relative to the camera's 3D reference frame, characterized by the 3 Euler angles $\{R^{xy}, R^{yz}, R^{zx}\}$. There is only one object model in each image. We also record the object's category $C^{cat}$ and the 3D model that has generated the image as the object's identity $C^{id}$.

### A.2  GENERATION OF DATASETS WITH REDUCED LATENT VARIABILITY

In addition to generating the full dataset with all the categories (TDW-117), we generated multiple datasets that contain the same number of images as TDW-117 (1.3 million images) with a reduced number of object categories. We call these TDW-N, where N represents the number of broader object categories used to generate the datasets. For each dataset, only object models belonging to the N classes are used to generate the dataset. For example, when N=2, the dataset contains only images from two object categories. For the analysis in Figure 2, Figure C.1, and Figure C.3, we used datasets where N = 2, 4, 6, 8, 16. For the analysis in Figure 5, Figure E.1, and Figure E.3, the reduced category variation dataset used only N = 1 category, and the full category variation dataset is the TDW-117 dataset.

We generated a dataset with reduced translation variation, and are used in the analysis of Figure 5, Figure D.4, Figure D.5, Figure D.6, Figure E.2, and Figure E.3. The full translation variation dataset is the TDW-117 dataset. For the reduced translation variation dataset, we generate a dataset with the same size and setting as in TDW-117, except that we set the camera to point at the object center before taking a picture of each object. So that the objects always appear at the center of each image, thus having translations very close to 0.

## B  MODEL ARCHITECTURE AND OPTIMIZATION DETAILS

### B.1  MODEL ARCHITECTURES

For analyses in Figure 2, Figure 3, Table C.1, Figure C.1, Figure C.2, Figure C.6, Figure C.7, and Table C.3, we used the ResNet-50 architecture. We found the neural alignment score results from ResNet-50 is qualitatively similar to results from ResNet-18 (Table C.2, Figure C.3), so we used ResNet-18 for subsequent analysis of neural representations in Figure 4, Figure 5, Figure C.4, Figure C.5, Appendix D and Appendix E. We implemented the model backbone using the PyTorch library (https://pytorch.org/vision/main/models/resnet.html). We changed the last linear layers of

the models to accommodate outputs for different tasks that we investigated. The architecture of ResNet-18 from which we extract activation to perform representational similarity analysis is shown Table B.1.

Table B.1: The architecture of ResNet-18. **Bold** font shows the layers from which we extract the output activation for analysis. We labeled these layers as "layer1.0.relu", "layer2.0.relu", "layer3.0.relu", and "layer4.0.relu" respectively. The activations are extracted from the ReLU layer after the activation from the skip connection is combined.

| Block name | Layer architecture | Layer position |
|---|---|---|
| | Conv (7x7, 64) → BatchNorm → ReLU | 1/18 |
| | MaxPool | |
| **layer1.0** | Conv (3x3, 64) → BatchNorm → ReLU | 2/18 |
| | Conv (3x3, 64) → BatchNorm → **ReLU** | **3/18** |
| | (skip connection combined before last ReLU) | |
| layer1.1 | Conv (3x3, 64) → BatchNorm → ReLU | 4/18 |
| | Conv (3x3, 64) → BatchNorm → ReLU | 5/18 |
| | (skip connection combined before last ReLU) | |
| **layer2.0** | Conv (3x3, 128) → BatchNorm → ReLU | 6/18 |
| | Conv (3x3, 128) → BatchNorm → **ReLU** | **7/18** |
| | (skip connection combined before last ReLU) | |
| layer2.1 | Conv (3x3, 128) → BatchNorm → ReLU | 8/18 |
| | Conv (3x3, 128) → BatchNorm → ReLU | 9/18 |
| | (skip connection combined before last ReLU) | |
| **layer3.0** | Conv (3x3, 256) → BatchNorm → ReLU | 10/18 |
| | Conv (3x3, 256) → BatchNorm → **ReLU** | **11/18** |
| | (skip connection combined before last ReLU) | |
| layer3.1 | Conv (3x3, 256) → BatchNorm → ReLU | 12/18 |
| | Conv (3x3, 256) → BatchNorm → ReLU | 13/18 |
| | (skip connection combined before last ReLU) | |
| **layer4.0** | Conv (3x3, 512) → BatchNorm → ReLU | 14/18 |
| | Conv (3x3, 512) → BatchNorm → **ReLU** | **15/18** |
| | (skip connection combined before last ReLU) | |
| layer4.1 | Conv (3x3, 512) → BatchNorm → ReLU | 16/18 |
| | Conv (3x3, 512) → BatchNorm → ReLU | 17/18 |
| | (skip connection combined before last ReLU) | |
| | AvgPool | |
| | Linear (fc) | 18/18 |

## B.2 MODEL TRAINING OBJECTIVES AND OTHER TRAINING DETAILS

We train models to estimate different subsets of the spatial or category latents conditioned on the image using supervised learning. For discrete targets like $\{C^{cat}, C^{id}\}$ in the latent set, we used the cross-entropy loss. The loss function for **object category classification** is given below. Suppose there are $N$ data samples in a mini-batch drawn from the dataset, which we denote as $\{(I_i, x_i, y_i, z_i, r_i^{xy}, r_i^{yz}, r_i^{zx}, c_i^{cat}, c_i^{id}) : i \in 1, 2...N\}$ where $I$ denote the image input. We parameterize the model as $f_\phi(I)$, the loss function is given by:

$$\mathbb{L}_{C^{cat}} = -\frac{1}{N} \sum_{i=1}^{N} \sum_{j=1}^{M} \log(f_\phi(C^{cat} = j|I_i)) \cdot 1\{j = c_i^{cat}\} \tag{1}$$

Where there are $M$ possible class for label $C^{cat}$, and $1\{j = c_i^{cat}\}$ is the indicator function that takes value 1 only when $j = c_i^{cat}$.

Similarly, the loss function for **object identity classification** is given below.

$$\mathbb{L}_{C^{id}} = -\frac{1}{N}\sum_{i=1}^{N}\sum_{j=1}^{M}\log(f_\phi(C^{id}=j|I_i))\cdot 1\{j=c_i^{id}\} \tag{2}$$

For continuous latents like $\{X, Y, Z, R^{xy}, R^{yz}, R^{zx}\}$, we used the square loss. The loss function for **distance regression** models is given below:

$$\mathbb{L}_{distance} = -\frac{1}{N}\sum_{i=1}^{N}(f_\phi(Z|I_i)-z_i)^2 \tag{3}$$

The loss function for **translation regression** models is the following:

$$\mathbb{L}_{translation} = \frac{1}{2}\Big[\mathbb{L}_x + \mathbb{L}_y\Big] \tag{4}$$

$$\mathbb{L}_x = -\frac{1}{N}\sum_{i=1}^{N}(f_\phi(X|I_i)-x_i)^2 \tag{5}$$

$$\mathbb{L}_y = -\frac{1}{N}\sum_{i=1}^{N}(f_\phi(Y|I_i)-y_i)^2 \tag{6}$$

For **rotation regression** task, we use the squared error between the model outputs and the $\sin$ and $\cos$ of the three Euler angles $\{R^{xy}, R^{yz}, R^{zx}\}$ characterizing the rotation of the object in the camera's 3D reference frame. For each Euler angle, the model has two output units $f_\phi(R_{\sin}^{xy}|I_i))$ and $f_\phi(R_{\cos}^{xy}|I_i))$ that are trained to match the $\sin$ and $\cos$ of the Euler angle respectively. The following is the loss function:

$$\mathbb{L}_{rotation} = \frac{1}{3}\Big[\mathbb{L}_{R^{xy}} + \mathbb{L}_{R^{yz}} + \mathbb{L}_{R^{zx}}\Big] \tag{7}$$

$$\mathbb{L}_{R^{xy}} = -\frac{1}{2N}\sum_{i=1}^{N}\Big[(\sin(f_\phi(R_{\sin}^{xy}|I_i))-\sin(r_i^{xy}))^2 + (\cos(f_\phi(R_{\cos}^{xy}|I_i))-\cos(r_i^{xy}))^2\Big] \tag{8}$$

$$\mathbb{L}_{R^{yz}} = -\frac{1}{2N}\sum_{i=1}^{N}\Big[(\sin(f_\phi(R_{\sin}^{yz}|I_i))-\sin(r_i^{yz}))^2 + (\cos(f_\phi(R_{\cos}^{yz}|I_i))-\cos(r_i^{yz}))^2\Big] \tag{9}$$

$$\mathbb{L}_{R^{zx}} = -\frac{1}{2N}\sum_{i=1}^{N}\Big[(\sin(f_\phi(R_{\sin}^{zx}|I_i))-\sin(r_i^{zx}))^2 + (\cos(f_\phi(R_{\cos}^{zx}|I_i))-\cos(r_i^{zx}))^2\Big] \tag{10}$$

When these objectives are combined, such as "Distance + Translation", we take the arithmetic mean of the loss functions of each task as the total loss. Loss function for **Distance + Translation**:

$$\mathbb{L}_{dis.+tra.} = \frac{1}{2}\Big[\mathbb{L}_{distance} + \mathbb{L}_{translation}\Big] \tag{11}$$

Loss function for **Distance + Translation + Rotation**:

$$\mathbb{L}_{dis.+tra.+rot.} = \frac{1}{3}\Big[\mathbb{L}_{distance} + \mathbb{L}_{translation} + \mathbb{L}_{rotation}\Big] \tag{12}$$

Loss function for **All classification + all regression**:

$$\mathbb{L}_{\text{All cla. + all reg.}} = \frac{1}{5}\Big[\mathbb{L}_{distance} + \mathbb{L}_{translation} + \mathbb{L}_{rotation} + \mathbb{L}_{C^{cat}} + \mathbb{L}_{C^{id}}\Big] \tag{13}$$

In addition to these models, we also trained models using the cross-entropy loss on the ImageNet dataset (Deng et al., 2009), which has 1000 classes. The untrained models we used are randomly initialized with different random seeds.

Before training, we normalized the continuous variables $\{X, Y, Z\}$ to have a standard deviation of 1 and centered around 0. We used mini-batch stochastic gradient descent with Adam optimizer (Kingma, 2014) with a learning rate of 0.001 for training the neural networks. Models are trained until they reach a plateau in test performance in a held-out test set. All experiments are trained with a batch size of 64. ResNet-50 models are trained with 1,000,000 batches, except those used for analysis in Figure C.2 are trained with 1,500,000 batches. ResNet-18 models are trained with 500,000 batches.

## C  VENTRAL-STREAM ALIGNMENT MEASURES

We use the benchmarks in the Brain-Score open science platform to measure the alignments of models trained with different tasks with the ventral stream neural and behavioral data. For most of our analyses in the main text, we used the set of public benchmarks. Specifically, we used "FreemanZiemba2013public.V1-pls" benchmark for V1 alignment, 'FreemanZiemba2013public.V2-pls' for V2 alignment, 'MajajHong2015public.V4-pls' for V4 alignment, and 'MajajHong2015public.IT-pls' for IT alignment. In addition, we got the models' behavioral alignment scores from the representation of the penultimate layers ('avgpool' layer) using the 'Rajalingham2018public-i2n' benchmark. We also evaluated a limited set of models on other non-public benchmarks on Brain-Score (see Figure C.5 and Figure C.6).

The neural benchmarks (V1, V2, V3, V4) measure how well a trained linear readout from the neural network models activation predicts the electrophysiological data in the respective brain regions. The procedure first identifies the best predictive layers for each region from a set of candidate layers in the model. The selected layer is then used to score against the data. In our evaluations using public benchmarks, we used the following candidate layers for ResNet-18 models: { 'relu', 'layer1.0.relu', 'layer1.1.relu', 'layer2.0.relu', 'layer2.1.relu', 'layer3.0.relu', 'layer3.1.relu', 'layer4.0.relu', 'layer4.1.relu', 'avgpool', 'fc' }. We used the following candidate layers for ResNet-50 models: {'relu', 'layer1.0.relu', 'layer1.1.relu', 'layer1.2.relu', 'layer2.0.relu', 'layer2.1.relu', 'layer2.2.relu', 'layer2.3.relu', 'layer3.0.relu', 'layer3.1.relu', 'layer3.2.relu', 'layer3.3.relu', 'layer3.4.relu', 'layer3.5.relu', 'layer4.0.relu', 'layer4.1.relu', 'layer4.2.relu', 'avgpool', 'fc' }. In our evaluations using non-public benchmarks, we used the following candidate layers for both ResNet-18 and ResNet-50 models: { 'conv1', 'layer1', 'layer2', 'layer3', 'layer4', 'fc' }. For behavioral scores, we used the activation from the penultimate layer 'avgpool' for scoring. Figure C.4 shows the layer assignments for ResNet-18 models in our experiments. For more details of the procedure on scoring, see Schrimpf et al. (2018) and https://www.brain-score.org/.

Table C.2 and Figure C.3 show the detailed region scores and behavioral scores of ResNet-18 models. Table C.1 and Figure C.1 show the detailed region scores and behavioral scores of ResNet-50 models. Figure C.2 shows how the mean neural alignment scores scale with the number of images we used to train those models.

Table C.1: Neural alignment scores of ResNet-50 models on Brain-Score public benchmarks. Each entry is the mean score of N=5 different models trained under the same condition with different random seeds. dis.: distance, tra.: translation, rot.: rotation.

| Training task | # output targets | V1 | V2 | V4 | IT | Neural mean | Behavior |
|---|---|---|---|---|---|---|---|
| Distance | 1 | 0.279 | 0.267 | 0.577 | 0.472 | 0.399 | 0.107 |
| Translation | 2 | 0.283 | 0.275 | 0.580 | 0.480 | 0.405 | 0.072 |
| Dis. + Tra. | 3 | 0.273 | 0.284 | 0.588 | 0.483 | 0.407 | 0.093 |
| Rotation | 6 | **0.302** | 0.283 | 0.568 | 0.460 | 0.403 | 0.171 |
| Dis. + Rot. | 7 | 0.284 | 0.272 | 0.571 | 0.481 | 0.402 | 0.212 |
| Tra. + Rot. | 8 | 0.276 | 0.271 | 0.580 | 0.484 | 0.403 | 0.141 |
| Dis. + Tra. + Rot. | 9 | 0.277 | 0.274 | 0.578 | 0.491 | 0.405 | 0.143 |
| Object category | 117 | 0.285 | 0.280 | 0.582 | 0.501 | 0.412 | 0.440 |
| Object identity | 548 | 0.271 | 0.278 | 0.588 | 0.509 | 0.412 | **0.443** |
| All spatial + classification | 674 | 0.262 | 0.289 | 0.586 | 0.506 | 0.411 | 0.413 |
| Untrained models | NA | 0.212 | 0.108 | 0.387 | 0.246 | 0.238 | 0.020 |
| ImageNet-1K | 1000 | 0.250 | **0.324** | **0.590** | **0.557** | **0.430** | 0.416 |

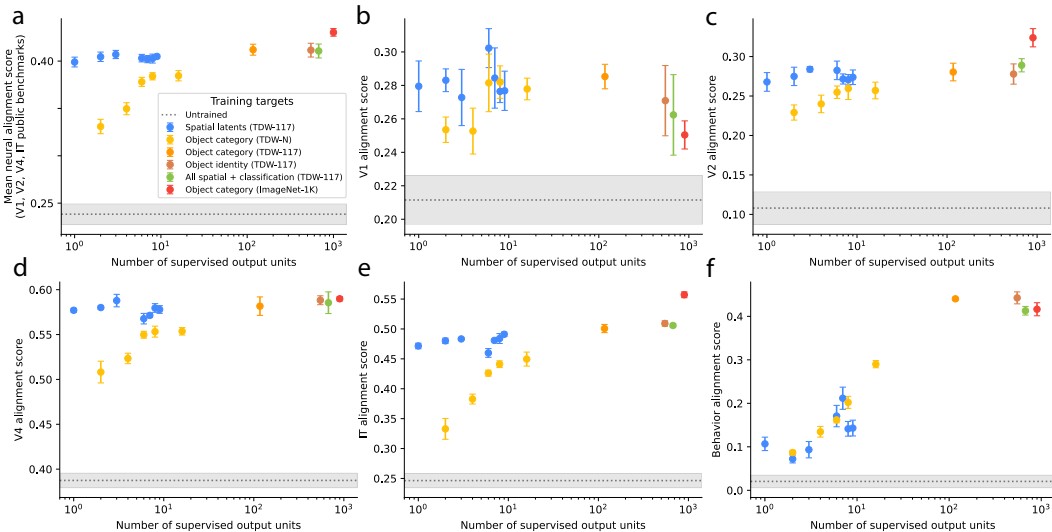

Figure C.1: Neural and behavioral alignment scores of ResNet-50 models on Brain-Score public benchmarks. **(a)** Mean neural score. **(b)** V1 alignment score. **(c)** V2 alignment score. **(d)** V4 alignment score. **(e)** IT alignment score. **(f)** Behavior alignment score. For results on other non-public benchmarks, see Figure C.6.

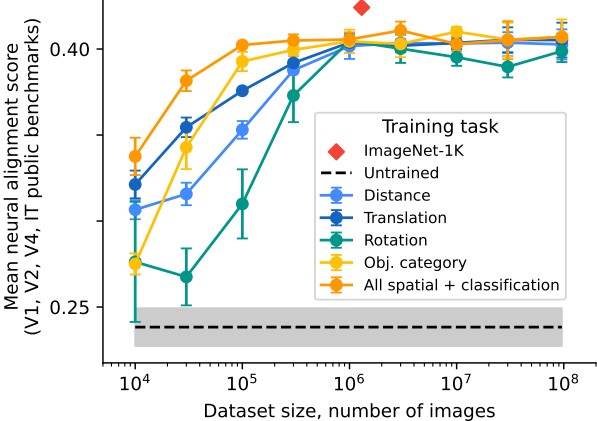

Figure C.2: Neural alignment (y-axis) of ResNet-50 models trained on different tasks as a function of the number of images in the image dataset (x-axis) used to train them. For all the tasks, the alignment scores initially scale with the dataset size but reach a plateau at around 1 million images. The alignment scores for models trained on different tasks are very similar when the training images are at or more than 1 million.

Table C.2: Neural alignment scores of ResNet-18 models on Brain-Score public benchmarks. The score is averaged over different models trained under the same condition with different random seeds (N=5 for untrained models, N=8 for the rest). dis.: distance, tra.: translation, rot.: rotation.

| Training task | # output targets | V1 | V2 | V4 | IT | Neural mean | Behavior |
|---|---|---|---|---|---|---|---|
| Distance | 1 | 0.264 | 0.248 | 0.562 | 0.448 | 0.381 | 0.109 |
| Translation | 2 | **0.266** | 0.258 | 0.560 | 0.455 | 0.384 | 0.048 |
| Dis. + Tra. | 3 | 0.260 | 0.257 | 0.563 | 0.458 | 0.384 | 0.034 |
| Rotation | 6 | 0.254 | 0.245 | 0.539 | 0.452 | 0.372 | 0.118 |
| Dis. + Rot. | 7 | 0.255 | 0.245 | 0.561 | 0.463 | 0.381 | 0.155 |
| Tra. + Rot. | 8 | 0.265 | 0.256 | 0.564 | 0.470 | 0.389 | 0.064 |
| Dis. + Tra. + Rot. | 9 | 0.265 | 0.259 | 0.566 | 0.468 | 0.389 | 0.078 |
| Object category | 117 | 0.248 | 0.266 | 0.583 | 0.492 | 0.397 | 0.352 |
| Object identity | 548 | 0.247 | 0.274 | 0.581 | 0.490 | 0.398 | 0.352 |
| All spatial + classification | 674 | 0.254 | 0.269 | **0.588** | 0.504 | 0.404 | **0.354** |
| Untrained models | NA | 0.217 | 0.114 | 0.381 | 0.246 | 0.239 | 0.026 |
| ImageNet-1K | 1000 | 0.250 | **0.307** | 0.584 | **0.512** | **0.413** | 0.327 |

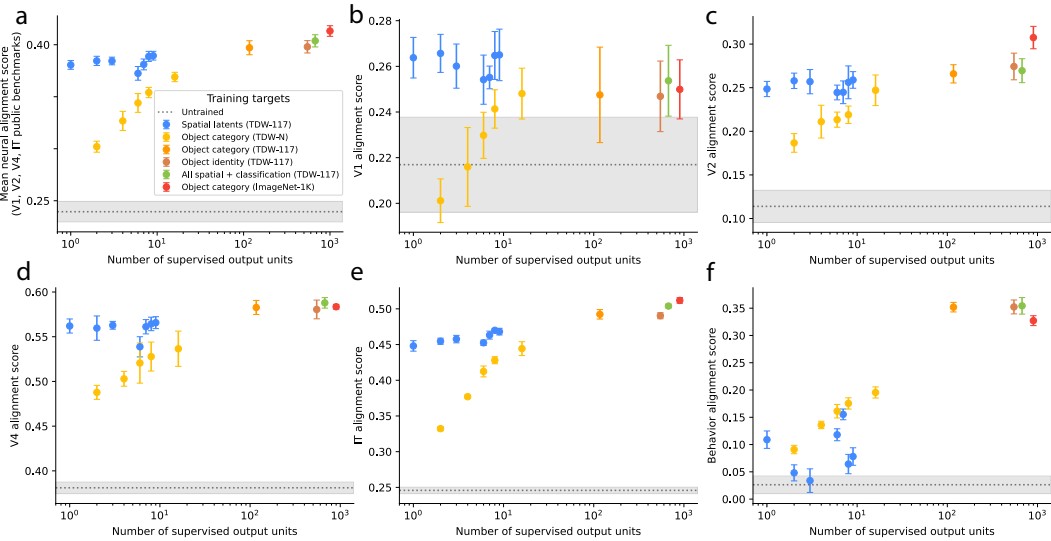

Figure C.3: Neural and behavioral alignment scores of ResNet-18 models on Brain-Score public benchmarks. **(a)** Mean neural score. **(b)** V1 alignment score. **(c)** V2 alignment score. **(d)** V4 alignment score. **(e)** IT alignment score. **(f)** Behavior alignment score. For results on other non-public benchmarks, see Figure C.5.

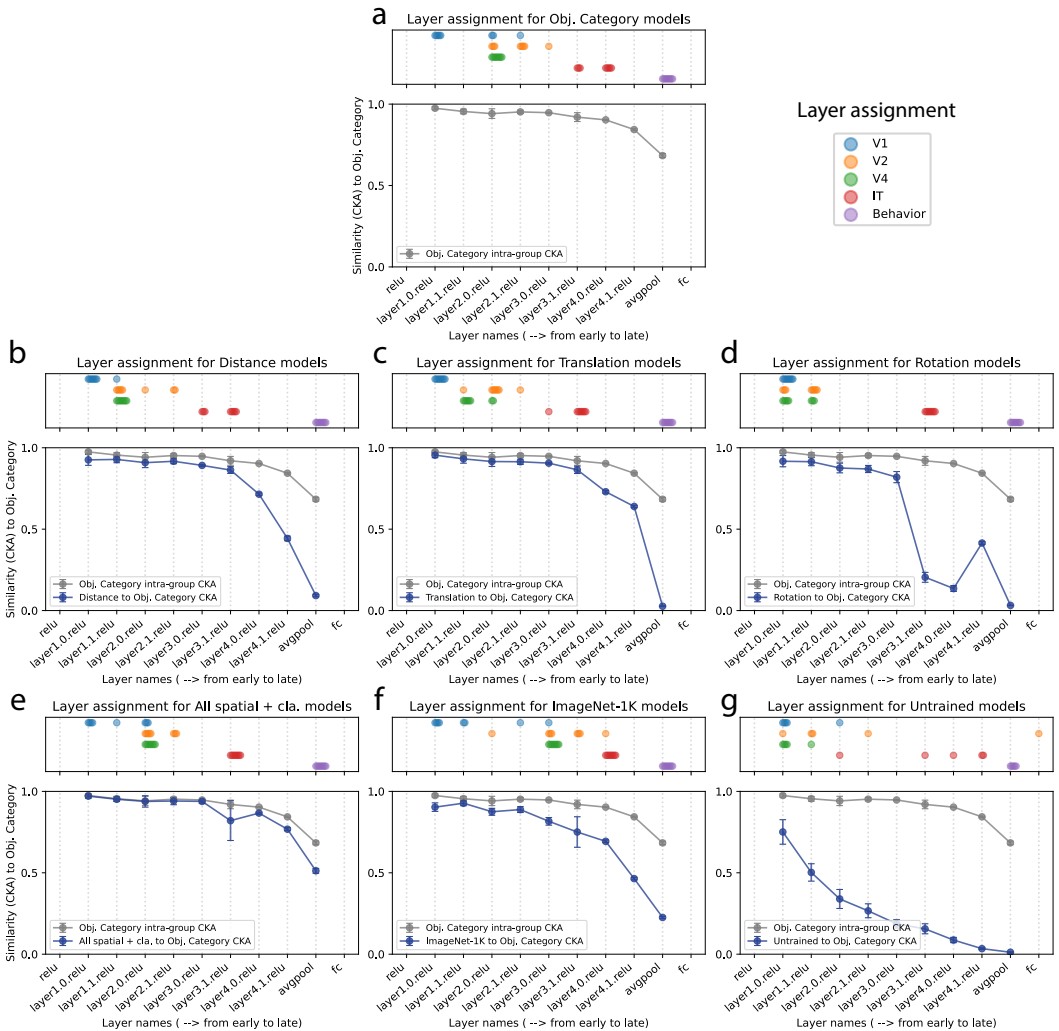

Figure C.4: Layer assignment details of ResNet-18 models trained on different tasks compared to their representational similarity to category-trained models. The dot positions indicate the best predictive layers for the regions on Brain-Score public benchmarks. The behavior readout layers are assigned to the penultimate layers by hand. **(a-g)** shows models trained on **(a)** object category, **(b)** distance, **(c)** translation, **(d)** rotation, **(e)** all categories and all spatial latents, **(f)** ImageNet-1K classification, and **(g)** untrained models. Most of the V1, V2, V4, and IT layers of latent-trained models are assigned to layers before their representations diverge significantly from the category-trained models. Compared to category-trained models, the layer assignments in spatial latent trained models are slightly shifted towards earlier layers.

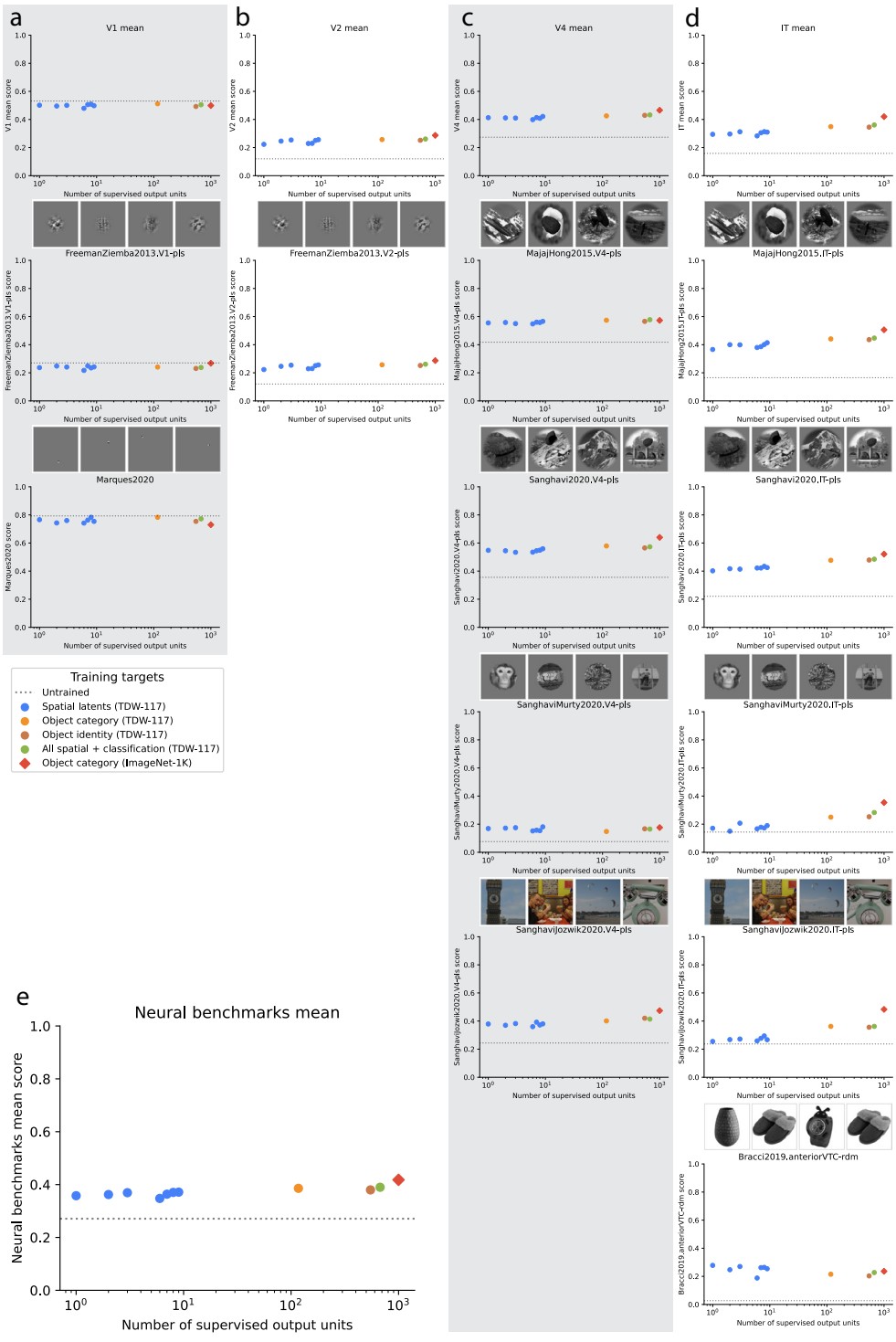

Figure C.5: Neural alignment scores of ResNet-18 models in other (non-public) neural benchmarks on Brain-Score. **(a-d)** show results for **(a)** V1, **(b)** V2, **(c)** V4, and **(d)** IT benchmarks. The top panel shows the mean score across all benchmarks in the region. Each point represents the result from one model. **(e)** shows the average of V1, V2, V4, and IT scores. Models trained on spatial latents performed comparably to category-trained models in most V1, V2, and V4 benchmarks. However, among the five IT benchmarks, spatial latent-trained models significantly underperformed category-trained models in two benchmarks ("SanghaviMurty2020.IT-pls", "SanghaviJozwik2020.IT-pls") while outperforming them in one ("Bracci2019.anteriorVTC-rdm").

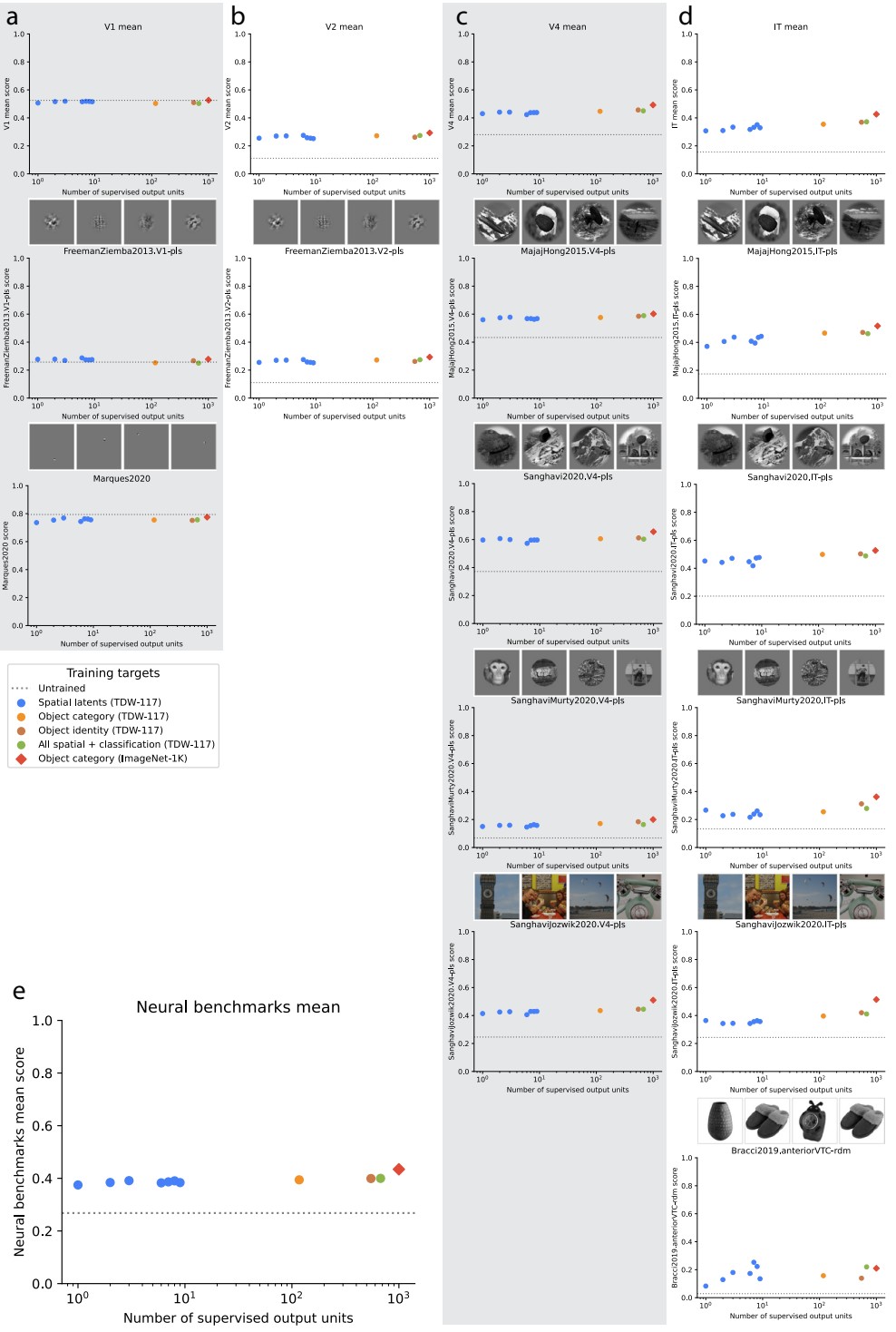

Figure C.6: Neural alignment scores of ResNet-50 models in other (non-public) neural benchmarks on Brain-Score. The layout is the same as Figure C.5

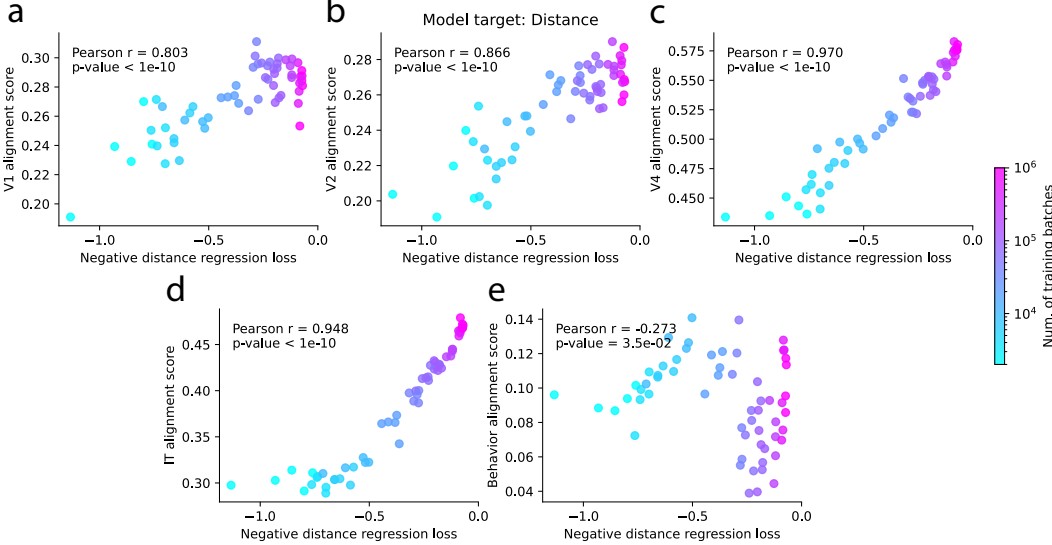

Figure C.7: A breakdown of neural alignment scores plotted against the overall distance regression performance of ResNet-50 models trained to estimate distance. **(a-e)** The alignment scores for **(a)** V1, **(b)** V2, **(c)** V4, **(d)** IT, and **(e)** behavior on Brain-Score public benchmarks. We see strong correlations between the distance regression performance with all neural alignment scores. The distance performance does not have a positive correlation with the behavioral alignment score. This figure shows results from multiple random initializations. Each dot shows a ResNet-50 model colored by the number of training batches. For model trained with other objectives, see Table C.3.

Table C.3: Correlations between model performance and neural/behavioral alignment scores of ResNet-50 models. The entries in the table show Pearson correlation coefficients between the model performance (negative losses on a held-out test set) and their neural/behavioral alignment scores on Brain-Score public benchmarks. For example, the "Translation" row shows the correlation between the translation regression losses and alignment scores from models trained on translation. The "Rotation" row shows the correlation between the rotation losses and alignment scores from models trained on rotation. There are strong positive correlations between all neural scores and all of the target performance. Distance and translation performance do not positively correlate with behavior scores, while rotation and classification tasks do.

| Model targets | V1 | V2 | V4 | IT | Neural mean | Behavior |
|---|---|---|---|---|---|---|
| Distance | 0.803 | 0.866 | 0.970 | 0.948 | 0.975 | -0.273 |
| Translation | 0.807 | 0.874 | 0.931 | 0.912 | 0.941 | -0.073 |
| Rotation | 0.659 | 0.723 | 0.706 | 0.785 | 0.736 | 0.744 |
| Object category | 0.735 | 0.947 | 0.945 | 0.969 | 0.958 | 0.982 |
| Object identity | 0.732 | 0.964 | 0.948 | 0.979 | 0.969 | 0.984 |

## D  SIMILARITY OF NEURAL REPRESENTATIONS

We use centered kernel alignment (CKA) to measure the similarity between the learned representations of different models (Kornblith et al., 2019). For most of our analysis, we extract the internal activations of our models from 4 different layers in ResNet-18 as in Table B.1. Those are activation of our models in response to 2000 held-out test images in the TDW-117 dataset that have full variations in all the latents we investigated. For all of our analysis, we used linear CKA. The implementation follows Kornblith et al. (2019).

For the analysis is Figure 4, we analyzed the pair-wise similarity between models trained with different objectives and models trained with the same objectives but initialized with different random seeds. To visualize the relationship between different groups, we used multi-dimensional scaling (MDS) to embed the model similarity matrix in a 2D space. The distance matrix used for MDS is calculated using the Euclidean distance between the row vectors of the similarity matrix between all of our models.

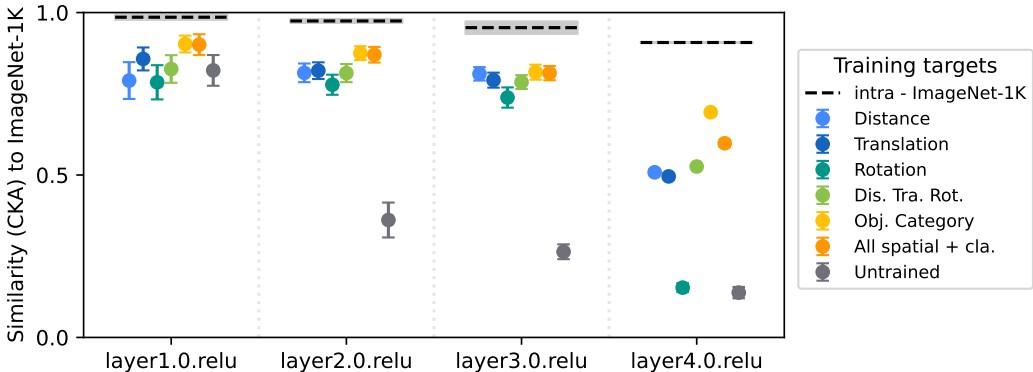

Figure D.1: The similarity of ResNet-18 models trained with different tasks to ImageNet classification (ImageNet cla.) models at different layers. intra – ImageNet cla. shows the averaged distance between categorization models trained with different random seeds.

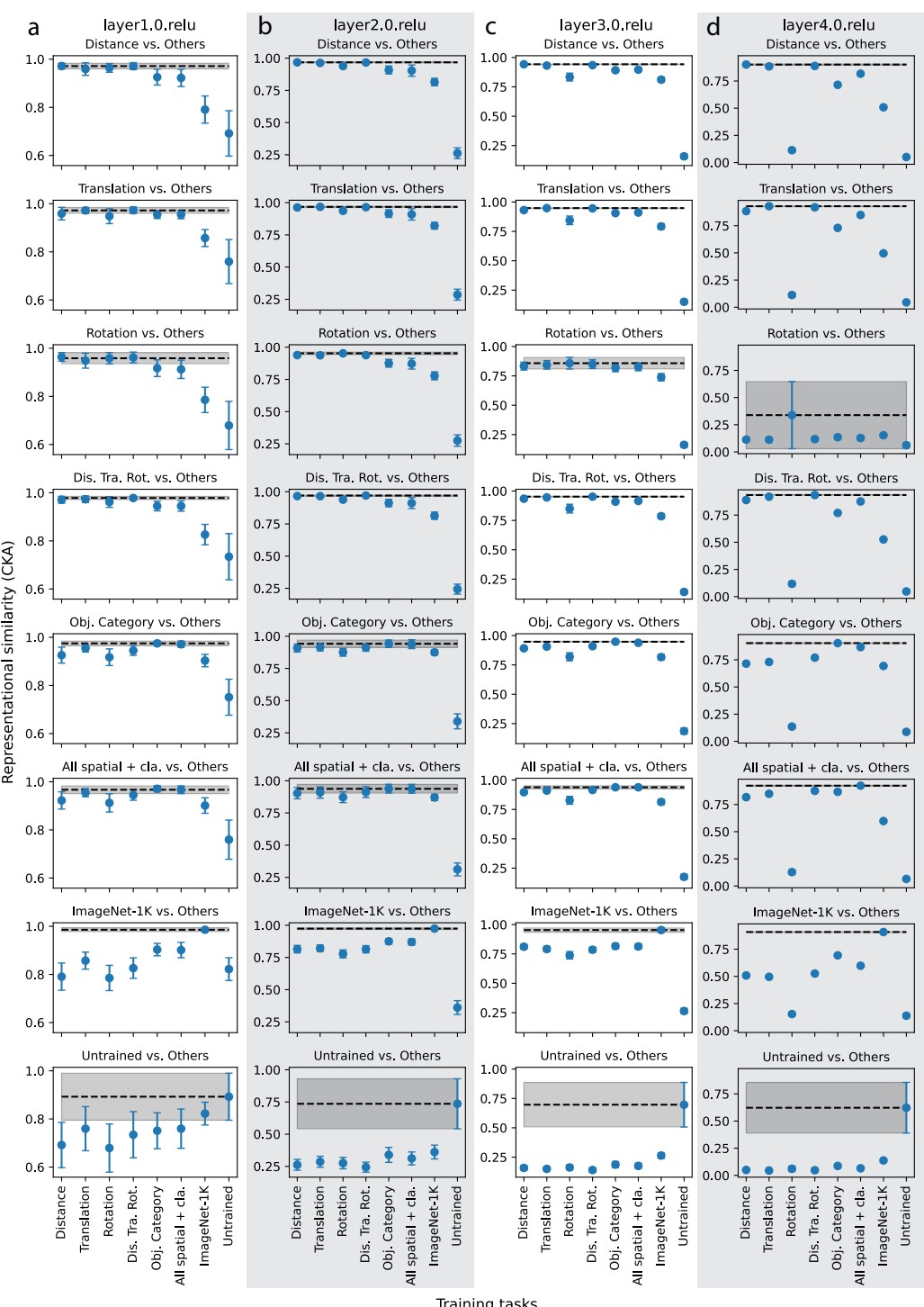

Figure D.2: Representational similarity (CKA) of ResNet-18 models trained on different tasks. Distance vs. Others means the CKA between models trained on distance regression and models trained on other tasks. Intra-group similarity measures the similarity of models within the same group and trained with different random initializations. Between-group similarity measures pairwise similarity of models in two different groups. In early to middle layers, most models trained to estimate different spatial latents are not much more dissimilar than models trained in the same group with different random seeds. Error bars or shaded regions show the SD of pair-wise model similarity. (a) early layer, layer1.0. (b, c) middle layers, layer2.0, layer3.0. (d) late layers, layer4.0.

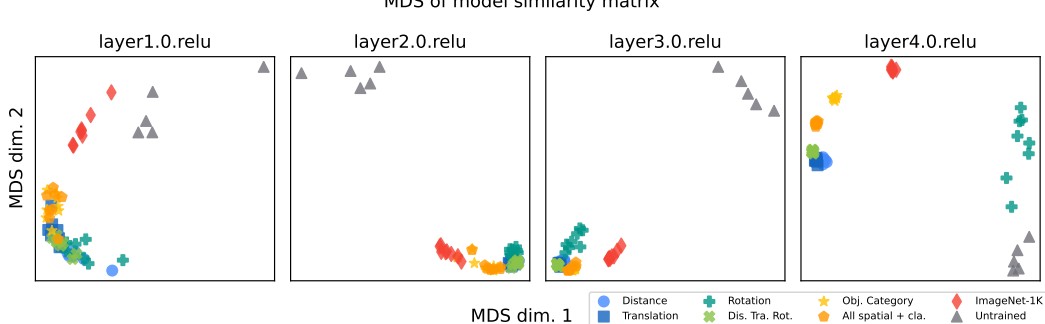

Figure D.3: Multi-dimensional scaling (MDS) visualization of similarity matrix of ResNet-18 models trained on different tasks. The representations of models trained on different spatial latents are mixed together until the last few layers (layer4.0.relu).

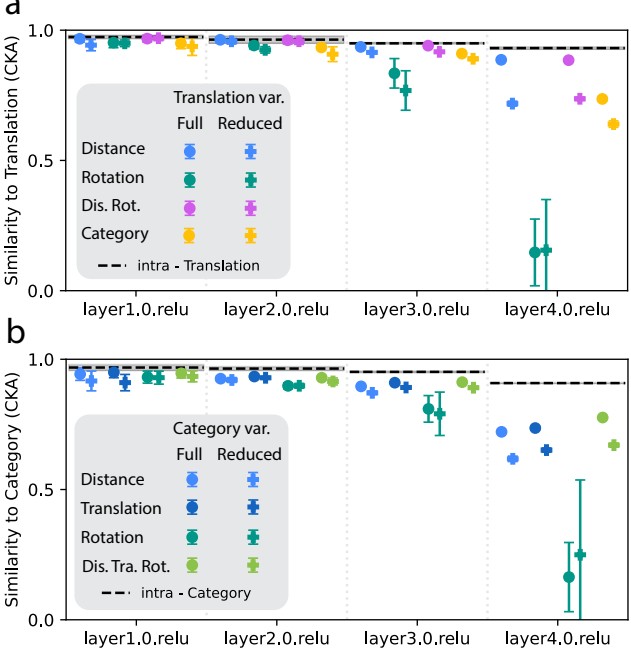

Figure D.4: Non-target latent variability helps models learn representations that are more similar to models trained directly on that non-target latent. **(a)** Similarity between translation-trained models (ResNet-18) and models trained on other targets. Although not trained to estimate translation, models trained on data with full translation variability learned representations that are more similar to translation-trained models than models trained with reduced translation variability. (We found this is true in most cases in layer3 and layer4, except rotation in layer4, which has a large SD) **(b)** Similarity between category-trained models and models trained on other targets. Although not trained to estimate category, models trained on data with full category variability learned representations that are more similar to category-trained models than models trained with reduced category variability. (We found this is true in most cases in layer3 and layer4, except rotation, which has a large SD). For both (a) and (b), error bars show the SD over CKA scores of 6x6=36 pairs of models.

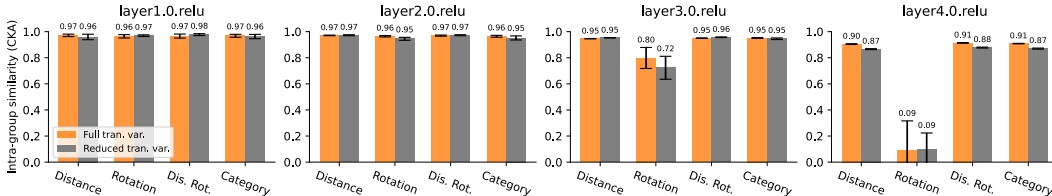

Figure D.5: The intra-group similarity between models (ResNet-18) trained to estimate different targets. The intra-group similarity is the mean pair-wise similarity between models trained on the same target but with different random initializations. Compared to models trained with reduced translation variability, models trained with full translation variability learned more similar representations in layer4, except rotation-trained models, which have large SD.

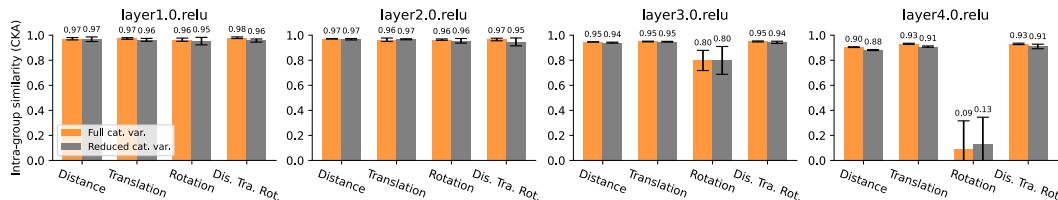

Figure D.6: Similar to Figure D.5, but compared to models (ResNet-18) trained with less category variability. Compared to models trained with reduced category variability, models trained with full category variability learned more similar representations in layer4, except rotation-trained models, which have large SD.

# E    ANALYSES OF REPRESENTATION OF NON-TARGET LATENTS

In Figure 5, Figure E.1, and Figure E.2, we analyzed the effect of non-target latent variability in the training data on the learned representations. Specifically, we investigated how well a model can represent non-target latents in their internal representations and whether more non-target latent variability helps learn better or invariant representations of them. We extracted activation from internal layers of models in response to the 2000 held-out TDW-117 test dataset that has variability in all latents we investigated. Many layers have very high dimensional activation, so we reduced the dimensions of the layers that are higher than 6515 dimensions to 6515 using random projection. This number 6515 is determined by the Johnson-Lindenstrauss lemma. We then use the dimensionally reduced representations to train mapping models that map from the activation to the specific non-target latents in the dataset. For discrete targets such as categories, we used linear support vector classification. For continuous targets, we used ridge regression. We run 5-fold cross-validation for each decoding layer-target pair and aggregate the decoding performance from multiple models initialized with different random seeds. The data shown in Figure 5, Figure E.1, and Figure E.2 is the mean of the aggregated decoding performance, and the error bars show the standard deviation of each aggregated condition.

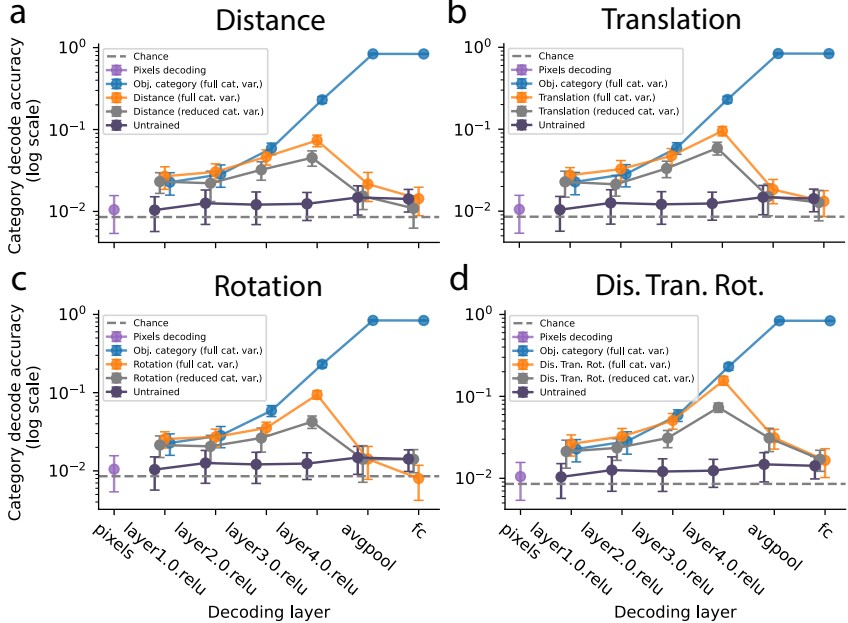

Figure E.1: Object category decoding performance at different layers of spatial latent-trained models (ResNet-18) compared with category trained and untrained models. (Orange curves show models trained to estimate: **(a)** distance, **(b)** translation, **(c)** rotation, and **(d)** distance + translation + rotation. The category decoding performance of spatial latent trained-models is similar to category trained models in early to middle layers. Then, they become divergent at late layers. Spatial latent-trained models usually have much better category decoding performance than untrained models. (cat. var. – category variability. Error bars show the SD across different cross-validation runs from different randomly initialized models.)

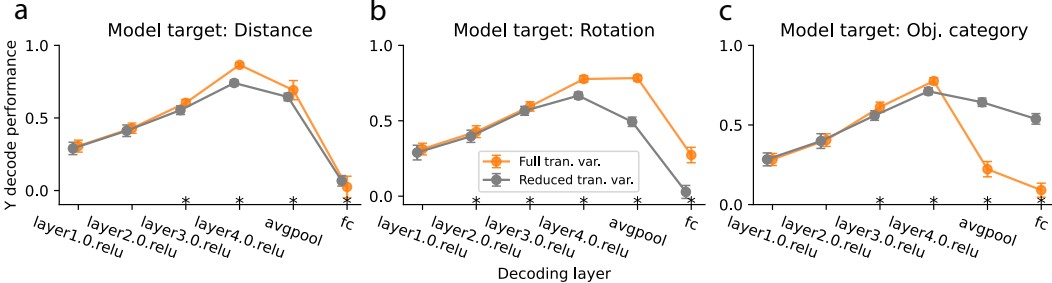

Figure E.2: Decoding performance for the non-target latent – translation, Y – when the models (ResNet-18) are trained to estimate target latents – **(a)** distance, **(b)** rotation, **(c)** category. Additional non-target latent variability helped these models learn better representations of the non-target latent in the intermediate layers. (tran. var. – translation variability. Error bars show the SD across 5 cross-validation runs × 6 randomly initialized models. "*" indicates a significant difference between the two groups, Mann-Whitney U test, p value < 0.05)

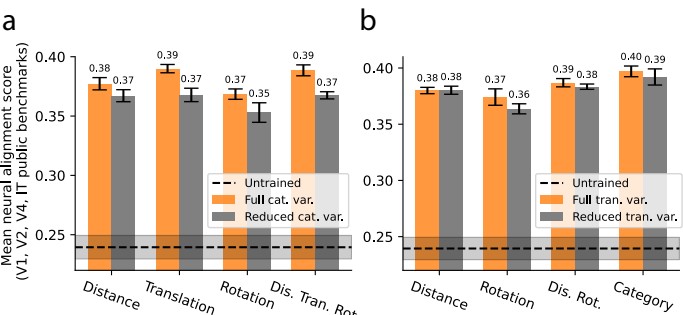

Figure E.3: Non-target variability helps models learn more neural-aligned representations. **(a)** Models (ResNet-18) trained to estimate different target latent with the full non-target (category) variations dataset have higher neural alignment scores than models trained with reduced non-target variation. **(b)** We see similar results when the non-target latent is translation and the targets are rotation or object category, but the effect is less pronounced.

