# OpenReview forum: "Vision CNNs trained to estimate spatial latents learned similar ventral-stream-aligned representations"
_ICLR.cc/2025/Conference — ICLR 2025 Poster_

### Official Review · Reviewer_wqcV · 2024-11-03

**Soundness:** 4
**Presentation:** 3
**Contribution:** 4
**Rating:** 8
**Confidence:** 5

**Summary:**

The paper assesses the neural predictivity and representational similarity of CNNs trained on a novel large synthetic dataset to decode different combinations of the following latent variables: object category , object identity, object position in 3-D space, and object orientation. The author's make the following observations:
- Training networks using a sufficiently large/realistic/diverse synthetic dataset can produce representations that rival those trained on natural images in terms of predictivity of ventral stream responses.
- Model's trained only to estimate spatial latents (i.e. without any explicit incentive for class invariance) can explain a similar fraction of variance as models trained to do categorization.
- The ability to decode spatial latents is correlated with predictivity.
- Model's trained on different tasks learn similar internal representations (as measured by CKA), especially at early layers and to a lesser degree deeper in the networks.
- The amount of variability of some latent variable in a training dataset modulates the extent to which said latent is linearly-decodable from a given representations, even when that latent is not used to train the neural network.

**Strengths:**

- The subject (the degree to which object classification is a unique normative explanation for features of neural representations in the ventral stream) is very interesting (and to a substantially large community).

- Investigating this hypothesis using a carefully controlled *but appropriately large* synthetic dataset is a sensible and novel approach.

- For the most parts experiments are clearly described, easy to follow, and sufficiently thorough to justify the conclusions.

- While I find the results of Section 3.3 the least convincing of those presented, doing these experiments is certainly a step toward addressing why similar representations are observed in the early layers across many artificial representations (which is "above and beyond" what is necessary for supporting the core arguments laid out in the paper).

**Weaknesses:**

- Layer Mapping Details Missing: I think it would be good to report which layers are mapped to each of the considered brain areas/datasets (unless I missed this detail). For instance, if the same layers are mapped to V1 across tasks this would be interesting especially in light of the fact earlier layers have higher CKA similarity.

- More fine-grained/controlled predictivity-to-decoding correlation analyses: It would be interesting to see if performance on any particular task is more or less correlated with predictivity in a particular region (i.e. by reproducing Fig. 3 but measuring the correlation with only V1 predictivity etc.). Furthermore, it may be easier to interpret this type of result if the layers used to decode the latent parameters were matched to the layers mapped to a particular region (so you can make statements along the lines of "the ability to decode distance from layer3.0.relu was correlated with the ability to predict V4 from the same activations, suggesting a possible functional link between the two").

- Difficult to draw conclusions from results in Section 3.3:
  - It would be nice to see the linear decoding performance for each latent on the pixels as well as early in the network as a baseline.
  - In Fig. 5b while all models have marginal increases  in category decoding (though most differences seem insignificant based on the depicted error bars, which are not described, but I assume to be 95% confidence intervals?), they are all still generally unable to decode categories at a meaningful rate. I think as well as the chance-performance floor it would be good to include a "trained for category decoding only" ceiling to help assess the scale of the improvement. Furthermore it would be good to include the performance curve (decoding accuracy as a function of depth in the network) for randomly initialized networks. If this curve is above the others a more accurate description would be "learning to decode a spatial latent decreases the decodability of a non-target latent less when that latent is more variable in the training set." Similar baselines would be useful in 5c.
  - The all vs. none variability setup in these experiments produced mostly small differences in the decoding performance with the notable exceptions of layer4.0relu in 5b bottom row and avgpool/fc in the bottom row of Fig 5c. Given this I think the result could be made more convincing by showing an orderly trend in decoding performance as the variability of non-target latents is parametrically decreased in steps between the Full variability case and the very low variability case explored here.  Finally in this setup it would be good to see how decoding performance varies in terms of the variability. Consider an experiment of this type:
    - You have (in addition to the models at full translation variability and zero translation variability already trained) a model trained using a new 50% translation variability dataset. You may then want to measure the translation decoding performance of all models both the full-variability and half-variability test-sets to get the clearest picture possible.
  - In sum, I think the language describing these results could be softened to reflect how difficult it is to draw definitive conclusions about internal representations from this type of experiment. This isn't a big issue at all but maybe changing line 430 to: "In summary, most of our observations are in line with the hypothesis that models develop better intermediate representations of nontarget latents in the presence of their variability, though future work is needed to understand both the size of and mechanism behind this effect."

- Model-Brain Comparisons Leave out Relevant Datasets: the author's only compare model responses to 1/4 of the neural recording datasets in both areas V4 and IT available on the BrainScore platform. I understand that predictivities across these datasets are correlated with each other, but I see no reason not to include them in the analyses. Especially given that the images presented in the Majaj-Hong benchmarks have a striking similarity in construction to the TDW synthetic dataset (object at random pose superimposed on a natural background), it would be good to confirm that the observations in this work generalize to other types of images!

**Questions:**

- About the spike in category decoding with rotation latent target (bottom row Fig. 5b): I wonder if rotation decoding is somewhat entangled with category decoding. I.e. to know some object has been rotated by 90 degrees you must know what its shape is/how it appears when presented at zero degrees of rotation (and such information is correlated with category). It might be interesting to look and see how the performance on category decoding breaks down by class (if category performance is high for objects whose shape varies strongly with viewing angle this might support such a hypothesis).

- Why are two different similarity metrics employed when comparing two artificial representations and when comparing artificial representations to neural measurements? Since the current work only considers the subset of datasets from BrainScore that are public, couldn't you use CKA to compare the artificial and biological representations as well (or alternatively use partial least squares regression to compare pairs of CNNs)? This doesn't seem likely to be an important issue but I am curious as to the reasoning, as it might be nice to have all comparisons on the same "scale", subject to the same method-induced biases, and tolerant to the same transformations.

- In the paragraph beginning on line 334 the authors suggest that their results from decoding/CKA experiments suggest that each model with similar predicitivity is also explaining the same portion of variance in the neural responses. While this seems sensible (especially perhaps for earlier layers/V1-V2 predictivity), I don't understand why we cannot just address this hypothesis directly. For example we can: directly examine the residuals produced by each model-to-brain prediction, predict the residual of one predictive model with another, or predict the neural data using ensembled models to get a sense for how overlapping the explained variances are. Again, given that the study is limited to the publicly available benchmarks all of these experiments seem feasible, and I think very interesting!

- Semi-Nitpick: the paper repeatedly references the number of units provided supervision. I don't think this is an interesting axis, or fair to compare between units that receive a discrete category label and a continuous latent variable. Meaning, you could discretize each spatial latent into 117 bins without changing the training signal but equalizing the number of outputs to the categorization task. I think rather than the dimensionality of supervisory signal the number of bits of supervision is the relevant thing to compare! I.e. since you know the distribution of locations you can measure the entropy of the translation variable and compare this to the entropy of the category variable (this may be trickier because of discrete vs. differential entropy etc.), but I think is more meaningful than comparing dimensionalities.

- Note: I would prefer to leave a score of 7 for this paper, but as that is unavailable I will sit at a soft 6 for now. I am very willing to raise my score if the authors can address/justify/explain some of the weaknesses or questions listed above.

---

> ### Author Response · Authors · 2024-11-13
>
> Hi reviewer wqcV, a huge thank you for giving such a thoughtful and detailed review and helping us improve this paper. We will address your first question in this response and work on addressing your other questions in a later response.
>
> **Response to your questions:**
> The ground-truth rotation of objects in our dataset used for training is defined relative to a canonical view of the object. The canonical view is tied to the object identity. Although there might be some simple heuristics to determine the canonical view of an unknown object, we believe that determining the canonical view mainly relies on correctly identifying the object first. So, as you pointed out, we also think that object rotation is entangled with object category, which may explain the spike in category decoding in rotation-trained models. The experiment you mentioned is a very interesting idea for testing this empirically; thanks for letting us know.
>
> We are working on addressing your other comments, so stay tuned. Thank you!

---

> ### Author Response · Authors · 2024-11-30
>
> We want to thank you for your effort in reviewing and helping us improve this paper.
>
> - Layer mapping details. Thanks for pointing that out. In the new pdf, we added a supplementary **figure C.4** to show the layer mapping details of our models. We also showed the layer mapping along the models’ CKA similarity to category-trained models. We found that many of the V1, V2, V4, and IT layers are mapped to layers before model representations start to diverge significantly from the category-trained models. V1 mapped layers are mostly consistent among models trained on different tasks. Compared to category-trained models, the V2, V4, and IT layers are usually shifted to earlier layers in spatial latent-trained models.
>
> - Fine-grained predictivity-to-performance correlation analyses. In our new pdf, we added a supplementary **figure C.6** and supplementary **table C.3**, which shows a more fine-grained analysis of how model performance in the trained latent task correlates with individual V1, V2, V4, IT, and behavior alignment scores separately. We found that all spatial latent estimation performance correlates strongly with all individual region neural scores. Distance and translation performance do not have a strong positive correlation with the behavior score, while rotation and category-trained models do. Our goal for the predictivity-to-performance analysis was to identify causally what training objectives (functional roles) could drive neural alignments if someone were to train a model, so we only analyzed the overall model performance on target latent estimation against the alignment scores and did not compare target/non-target latent decoding performance with neural alignment scores. We realized that our writing might not have been very clear in the previous version, which could cause confusion, so we revised some wording in the figure 3 captions and the main text. We think the idea of correlating the decoding performance of target/non-target spatial latents with neural alignment in the mapped layer is an interesting idea and could provide additional insights. However, we believe that conducting the proposed experiment falls outside the scope of the core claims of our current work. We leave this question to future investigations and hope our findings presented here will serve as a starting point for exploring such ideas.
>
> - Results in section 3.3: Statistical significance of our results. We remade the figures in section 3.3 (**figure 5** and others). The new version showed results from models trained with the same conditions as models investigated in section 3.2 (results from the previous version were from models trained with fewer batches). We found qualitatively similar results to those in the previous version, and the results in the new version are more significant due to longer training. We added a description in the captions of **figure 5** about the error bars and indicated the statistical significance of the differences between the two groups in the figures using “*” marks. From these analyses, we are confident that the differences we described in the main text were significant.
>
> - Results in section 3.3: Decoding performance. To help assess the scale of the improvement in decoding categories, we added a supplementary **figure E.1**, in which we showed the category decoding performance from spatial latent-trained models compared to category-trained models, as well as pixels and untrained models. We found that the category decoding performance of spatial latent-trained models is better than that of untrained models. So, spatial latent training increases category decoding performance in intermediate layers instead of decreasing it. Meanwhile, category decoding performance from these spatial latent-trained models is similar to category-trained models in the early to middle layers, but they start to diverge in late layers. Spatial latent-trained models become much worse at representing categories than category-trained models in late layers. This is consistent with our findings in Figure 4, which shows that the representations of spatial-latent trained models are very similar to category-trained models in early and middle layers and diverge in late layers.

---

> ### Author Response · Authors · 2024-11-30
>
> - Results in section 3.3: Interpolating non-target latent variability. Thank you for proposing the intriguing idea of measuring the impact of latent variability by interpolating between full and very low variability cases. While this more detailed analysis could indeed provide additional insights, we believe our core claims still hold even without these experiments. Furthermore, the analyses in our revised version showed the statistical significance of our results, which we hope addresses some of your previous concerns. Conducting such an experiment would require generating new datasets and retraining models, which, unfortunately, was not feasible given the computational and time constraints of the revision period. We acknowledge the value of this idea and will leave it as a direction for future work.
>
> - Results in section 3.3: Revise the language in line 430. Thanks for making the suggestion about revising the language describing our results in section 3.3. We revised the conclusion sentence based on your suggestion (now in **line 432**).
>
> - Model-brain comparisons on more datasets. Our neural alignment results in the main text are primarily based on the public benchmarks on Brain-Score. We acknowledge that the Majaj-Hong benchmarks are based on images with a lot of similarity to TDW images we used to train the models since they all involve synthetic objects superimposed on random backgrounds. In the revision, we further evaluated our models on other non-public benchmarks in Brain-Score, many of which contain images that are out of the training distributions of our models, such as more complex natural images. These results were only available to us recently due to previous technical issues in obtaining model alignment scores on these non-public benchmarks in Brain-Score. We evaluated our ResNet-18 models and showed the results from 12 out of the 16 available high-level neural benchmarks in supplementary **figure C.5**. We excluded the results from the four tong.Coggan2024 benchmarks because they give much lower scores for ImageNet-trained models than untrained models, which is in contrast with most of the other benchmarks in Brain-Score – we are unsure whether they are implemented correctly. In the rest of the 12 benchmarks, we found that models trained on spatial latents performed comparably to category-trained models in most V1, V2, and V4 benchmarks. However, among the five IT benchmarks, spatial latent-trained models significantly underperformed category-trained models in two benchmarks (”SanghaviMurty2020.IT-pls”, ”SanghaviJozwik2020.IT-pls”) while outperforming them in one (”Bracci2019.anteriorVTC-rdm”). This results in a lower averaged IT alignment score for most of the latent-trained ResNet-18 models than category-trained ResNet-18 models if we simply average across the full set of non-public benchmarks. These three benchmarks (”SanghaviMurty2020.IT-pls”, ”SanghaviJozwik2020.IT-pls”, and ”Bracci2019.anteriorVTC-rdm”) are based on natural images that are out of the training distribution of our models. We think these results invite future research to determine the extent to which our results can be generalized to many other out-of-distribution images. We added this to the discussion of the limitations and future directions at the end of the main text in **line 528**.

---

> ### Author Response · Authors · 2024-11-30
>
> - Why use regression-based metrics for brain-model comparison and CKA for model-model comparison? We used different metrics because we wanted to answer two different questions, which are traditionally studied in two different but related fields with their own methodological traditions. To assess how well spatial latent-trained models align with neural responses, we used Brain-Score, which incorporates many benchmarks based on regression-based metrics. These metrics are widely used in neuroscience not only for evaluating brain-model alignment but also for providing insights into how well neural activity can be predicted using model features. This is directly relevant to applications such as population neural control (Bashivan, Kar, & DiCarlo, 2019). We used CKA to compare model-model similarities because we want to answer the question: How different, if at all, are the representations learned by models trained on different spatial latents? CKA is one of the most popular approaches to compare different neural network model representations, mostly due to its thoughtful consideration of the invariant properties of similarity metric and its ability to reveal similarity between models trained with different random initialization. When designing CKA, the authors considered properties such as the sensitivity of neural network training to certain representational transformations (Kornblith, et al., 2019), which may not be relevant to real neural data. The decision to use regression-based metrics for brain-model comparisons and CKA for model-model comparisons reflects the origins of these methods in different fields that try to answer slightly different questions. Since our paper addresses both questions, we want to prioritize using the widely adopted approach so that our results can connect and remain comparable to previous studies. While it is possible to use CKA for brain-model comparisons or regression-based metrics for model-model comparisons, interpreting such results in the context of existing literature would be challenging. Moreover, brain data often differs from model data. Unlike artificial units in models, each unit response in the brain is typically an average of multiple repetitions, and the variability across these repetitions must be accounted for. That said, we appreciate the reviewer’s comment that it would be nice to have “all comparisons on the same ‘scale,’ subject to the same method-induced biases.” We acknowledge that alternative methods, such as the RSA framework, could potentially enable a more unified comparison between model-model and model-brain analyses (Schütt et al., 2023; Diedrichsen et al., 2020). While incorporating such an approach would undoubtedly add value, we believe it falls outside the scope of the current work and would not impact our core claims. We have included this point in the discussion of limitations and future directions at the end of the main text (**line 535**). Due to the limited time available during the revision period, we had to leave this as an avenue for future exploration.
>
> - Determine whether different models explain the same or different portions of variance in neural data. We thank the reviewer for raising this very interesting question. We did not address this question directly because, at the time of writing the paper, we were not aware of a thorough and robust method to conduct the analysis required to answer it. Also, we think that our core claims in the paper are supported by other experiments, and this question, although interesting, seems out of scope. We are grateful for the suggestion to explore methods such as predicting regression residuals or using ensemble models. However, given the limited time available, we must leave this as an avenue for future research. Thank you for highlighting this valuable direction.
>
> References:
> 1. Bashivan, P., Kar, K. and DiCarlo, J.J., 2019. Neural population control via deep image synthesis. Science, 364(6439), p.eaav9436.
> 2. Kornblith, S., Norouzi, M., Lee, H. and Hinton, G., 2019, May. Similarity of neural network representations revisited. In International conference on machine learning (pp. 3519-3529). PMLR.
> 3. Diedrichsen, J., Berlot, E., Mur, M., Schütt, H.H., Shahbazi, M. and Kriegeskorte, N., 2020. Comparing representational geometries using whitened unbiased-distance-matrix similarity. arXiv preprint arXiv:2007.02789.
> 4. Schütt, H.H., Kipnis, A.D., Diedrichsen, J. and Kriegeskorte, N., 2023. Statistical inference on representational geometries. Elife, 12, p.e82566.

---

> ### Author Response · Authors · 2024-11-30
>
> - Dimensionality of the supervisory signal vs entropy. We appreciate the reviewer raising this insightful point. Ideally, we would like to measure how “strong” the supervisory signals provided to the models during training are. However, at the time of writing this paper, we were not aware of a robust measure for this, and we believe our core claims remain unaffected by the specific measure chosen. We acknowledge that using dimensionality alone has its limitations. However, using entropy as a measure has challenges as well. For instance, entropy may be meaningful when comparing distributions with the same support, but comparing continuous entropy with discrete entropy poses difficulties. Discrete entropy is always non-negative, whereas continuous entropy can take negative values. Additionally, continuous entropy is sensitive to the scale of the supervisory signal. For example, the continuous entropy of a Gaussian distribution is given by:
> $$ \frac{1}{2} \log{2\pi \sigma^2} + \frac{1}{2}$$
> This value depends solely on the spread of the target distribution. In practice, however, when training machine learning models, targets are often scaled to a reasonable range, and our intuition suggests that this scaling should not significantly affect the “strength” of the supervisory signal. Yet, such scaling can dramatically alter the computed entropy of the target distribution. We therefore chose to use the dimensionality of the supervisory signal as an intuitive way to visualize and present our results, rather than as a rigorous metric for comparing the efficiency of these models in utilizing supervisory signals. We did not claim that latent-trained models are superior to category-trained models in this regard. Our intention was to highlight that it is possible to achieve comparable models using supervision based on a few latent variables. If there are better ways to measure the desired properties, we are open to suggestions and welcome further input. Thank you again for your thoughtful feedback.
>
> We hope our response answers your questions and addresses your concerns. If you have any other questions, please let us know!

---

> > ### Comment · Reviewer_wqcV · 2024-12-03
> > **Thank you!**
> >
> > Thank you to the author's for their efforts to address the questions raised in my review! I believe that the additional details, analyses, and discussion of the results (i.e. for predictivity on more diverse datasets) strengthen the contribution and make it considerably easier for the community to understand the implications of this (interesting) study in the context of the broader literature.
> >
> > Re: number of latent variables, I agree that the strength of the supervisory signal is difficult to quantify meaningfully. I guess my point is really just that I view category as a single latent variable (that for convenience purposes is fed to the model as a higher-dimensional but 1-hot vector). So, the interesting result is not that a "small number of latents" can lead to predictive representation but simply that non-category latent prediction can lead to similar representations (as we discussed above there are a variety of possible reasons for this, but providing a mechanistic explanation is somewhat out of scope for this conference paper). On other issues raised in the response I am generally in agreement, and I hope the author's consider adding similar language to the text/appendix where relevant in the final version!
> >
> > As a final suggestion, I believe making the entire TDW dataset available for download would in the long run strengthen the impact of this work. Making the code for generating the dataset available is a step in the right direction, but lowering the activation energy for future experimenters would likely be appreciated by many.
> >
> > I am raising my score to reflect these updates.

---

> > > ### Author Response · Authors · 2024-12-04
> > >
> > > Thank you again for reviewing and providing these additional very helpful suggestions; we really appreciate it!

---

### Official Review · Reviewer_acUt · 2024-11-04

**Soundness:** 4
**Presentation:** 3
**Contribution:** 3
**Rating:** 6
**Confidence:** 4

**Summary:**

The paper examines the effect of training models to estimate spatial latents on neural alignment (the "where"), comparing this approach with object recognition (the "what"). It highlights that to achieve high neural alignment, training with a focus on "where" yields similar benefits compared to the more common focus on "what." Interestingly, with fewer presentations in the "where" case, alignment levels appear comparable, and representations in the models' initial layers are strikingly similar.

**Strengths:**

The paper addresses an interesting problem that has been overlooked for some time. The presentation is highly effective, and the paper is easy to read. It takes an unconventional approach to the problem by focusing on spatial location, which proves to be very effective even in a low data regime. There are several controls in the main text and supplementary material that support the claims.

**Weaknesses:**

I know that in  semantic segmentation and other tasks where location and classification of the object is used for training, has been studied for brain-score but have not really succeeded.  Perhaps worth adding a bit of discussion on this topic, because even when its possible to say that the world is 3D and therefore the tasks seem to align better with vision, is also true that the models used in this work are only receiving 2D images as input. Some specific questions:

1. How this approach differs from previous attempts using semantic segmentation for brain-score alignment?
2. Why this  spatial latent estimation tasks succeed where others have not
3. Are there any implications of using  2D inputs to estimate 3D spatial properties, how this  relates to biological vision?

Since the title starts with  "Vision Models...", I largely missed other models like ViTs  to be studied. But it seems that as it stands the results only cover CNNs?
1. I would suggest  a change of the title if that is the case, or inclusion of other kind of vision models in the results to make sure is align with the title.

**Questions:**

Left in weakness section.

---

> ### Author Response · Authors · 2024-11-13
>
> Hi, reviewer acUt, thank you for your reviews and efforts in helping us improve the paper!
>
> **Response to your questions:**
> We know some work, such as Conwell et al. 2022, that includes semantic segmentation models. We would appreciate it if you could point us to some other studies that you think are relevant to the neural alignment of semantic segmentation models.
>
> The models we studied in this paper primarily focus on estimating object-centric spatial latent variables, including distance, translation, and rotation. We chose to examine models estimating these latents for three main reasons: (1) these variables constitute the core elements of the spatial information about objects, which people can intuitively estimate; (2) previous electrophysiological studies suggest that the ventral stream encodes these spatial attributes (Hong et al., 2016); and (3) models that predict these latents can be trained on the same dataset with the same architecture, allowing us to isolate the effect of the training objective while controlling for other factors.
>
> Most segmentation models we know require an additional decoder module to output a segmentation map, making their architecture different from the latent estimation models studied here. Our study aimed to analyze the influence of training objectives on learned representations while keeping the dataset and architecture constant. We didn't study the segmentation models here because it is hard to control their architecture to be the same. We also don't know whether segmentation models have higher or lower neural alignment. The TDW dataset we proposed can also contain ground-truth labels for segmentation maps, which could be a resource for people interested in studying segmentation models in the future.
>
> We are working on addressing your other comments, so stay tuned. Thank you!
>
> Reference:
> 1. Conwell, C., Prince, J.S., Kay, K.N., Alvarez, G.A. and Konkle, T., 2022. What can 1.8 billion regressions tell us about the pressures shaping high-level visual representation in brains and machines?. BioRxiv, pp.2022-03.
> 2. Hong, H., Yamins, D.L., Majaj, N.J. and DiCarlo, J.J., 2016. Explicit information for category-orthogonal object properties increases along the ventral stream. Nature neuroscience, 19(4), pp.613-622.

---

> > ### Comment · Reviewer_acUt · 2024-11-26
> >
> > Thank you for providing more context. According to the comment there was something else coming?  just wondering if there is something to add, before the discussion period ends. Looking forward.

---

> > > ### Author Response · Authors · 2024-11-26
> > >
> > > Yes, that is correct. We are working on an updated pdf with other responses. Stay tuned!

---

> ### Author Response · Authors · 2024-11-30
>
> We want to thank you for your effort in reviewing and helping us improve this paper.
>
> - Thank you for your suggestions about the paper title. We realized that using “vision models” in our title might be confusing since it makes people think about vision transformers as well. Our current study focuses on CNNs. So, in the new pdf, we changed the title from “vision models” to “vision CNNs.”
>
> - Biological vision and estimating 3D spatial latents from 2D input. In biological vision, humans and animals can extract significant 3D information from 2D visual cues alone. For example, humans rely on various cues in depth estimation, including binocular disparity, motion parallax, relative object size, shading, texture gradients, and occlusion. Many of these cues, such as relative size, shading, texture gradients, and occlusion, are readily available in 2D images. The biological vision systems are likely wired to exploit these cues for 3D spatial estimations from 2D images. Our results showed that it is possible to train models to estimate 3D spatial latents from 2D images and that these models achieve neural alignment with the ventral stream comparable to category-trained models. We think the implications of these results are that we should not assume the ventral stream is optimized for object recognition only, as optimizing for spatial tasks could also improve neural alignment with the ventral stream.
>
> - The reviewer has raised a very intriguing question: is it specifically the 3D spatial properties that drive this neural alignment, or could it also be explained by models trained solely on 2D property estimation? This remains an open question, and it is unclear whether models trained to estimate purely 2D spatial properties would outperform or underperform compared to our current 3D spatial latent-trained models. Answering this question would require controlled experiments to compare models trained on 3D spatial tasks with models trained on 2D spatial tasks, such as semantic segmentation tasks you mentioned. While this is a fascinating avenue for further exploration, it falls outside the scope of our core claims in the present study. We leave this question to future research. We also hope that our datasets and current results provide a valuable resource for future investigations on this question.
>
> We hope our response answers your questions and addresses your concerns. If you have any other questions, please let us know!

---

### Official Review · Reviewer_nMg4 · 2024-11-04

**Soundness:** 3
**Presentation:** 3
**Contribution:** 3
**Rating:** 8
**Confidence:** 4

**Summary:**

The paper asks whether the ventral stream only codes object category information. By showing that CNN models that are trained for spatial understanding can also align with the neural reponse in the ventral stream, the author provide a critical reflection on the assumption that ventral stream in the brain primarily encode categorical information. The authors also introduce synthetic datasets generated by a 3D graphics engine (ThreeDWorld) containing precise spatial labels, overcoming the scarcity of natural datasets with spatial information. This dataset enabled systematic experiments to evaluate CNNs trained on spatial latents.

**Strengths:**

* The paper asks an important question of the function role of the ventral stream besides image categorization. The discovery suggests that representation that is similar to the ventral stream neural response can be derived from training the network under spatial latent estimation objective as well. This contrasts the traditional belief that the primilary goal of the ventral stream is for object categorization only.
* To demonstrate the proposal, the paper utilizes a 3D graphics engine to generate large dataset to tackle the shortage of spatial latent variables in previous studies. In Figure C.3 in supplementary, the paper demonstrates the importance of having larger scale dataset.
* The paper shows that CNN trained on purely synthetic dataset can achieve a similar neural alignment score as these trained with natural images.
* The paper has clear writing and presentations.

**Weaknesses:**

* It could be more comprehensive if the paper utilizes natural images as training, e.g. one could extract sptial variables from SA-1B segmentation dataset to perform the experiments.
* The evaluation of the neural alignment score is limited to the Brain-Score. More metrics would be helpful to consolidate the results.
* The idea that ventral pathway is not responsible for classification alone is not new among cognitive neuroscience community. E.g Connor discussed in his paper that the retinotropic spatial information is transformed into relative positional information across ventral stream [1]. A discussion with related works in neuroscience community would be appropriated.

[1] Connor, Charles E., and James J. Knierim. "Integration of objects and space in perception and memory." Nature neuroscience 20.11 (2017): 1493-1503.

**Questions:**

* In Figure 5b, the decoding accuracy for classification is quite low for models trained for spatial latents, suggesting that the model trained for spatial latents is not suitable for classification. However, this contrasts with the CKA analysis in Figure 4b where the representation similarity is quite high between models trained for spatial latents and models for object categorization. How would you explain such difference?

---

> ### Author Response · Authors · 2024-11-13
>
> Thank you very much for your thoughtful reviews and help in improving the paper!
>
> **Response to your questions:**
> Thanks for pointing that out. We believe a couple of reasons could underlie the phenomenon you mentioned.
> 1. Figure 4b shows the CKA results until layer 4.0, an intermediate layer in the CNN (the 15th layer of the 18-layer ResNet). To decode categories from intermediate layers linearly could be challenging even for category-trained models. So, it is possible that even for category-trained models, the decoding performance will not be much better than the decoding performance from spatial latent trained models shown in Figure 5b at layer 4.0. However, this remains unknown; we can do an experiment to address that.
> 2. It is still possible that representations can have different category decoding performances while having similar CKA since CKA does not capture every aspect of neural representations.
>
> We are working on addressing your other comments, so stay tuned. Thank you!

---

> ### Author Response · Authors · 2024-11-29
>
> We want to thank you for your effort in reviewing and helping us improve this paper.
>
> - Category decoding. First, we want to expand on our previous response to your question about category decoding results. In the revised pdf, we included a new supplementary **figure E.1**, showing how well we can decode categories from spatial latent-trained models compared to category-trained models. We found that the category decoding performance of spatial latent-trained models is similar to category-trained models in early to middle layers (until layer3.0.relu); even the category-trained models have very limited ability to decode categories in these layers. The decoding performance starts to diverge in late layers (layer4.0.relu). This is also the point where we see a significant divergence in the representation similarity, accessed by CKA, in Figure 4. In the last few layers, spatial latent-trained models become much worse at representing categories than category-trained models. We think the finding in figure E.1 is consistent with our findings in figure 4.
>
> - Thank you for pointing us to the very interesting review paper by Connor and Knierim. In our new pdf, we cited this paper in the introduction section (**line 48**). We also revised our introduction section, which includes discussions and references to previous works emphasizing that the ventral stream is not responsible for classification alone.
>
> - Natural image training and additional evaluation metrics. Our current study focuses on a detailed examination of many object-centric spatial latents in 3D space and their impact on the learned representations. We wanted to isolate the effects of different training objectives, so we deliberately kept the dataset and model architecture constant. Currently, we are not aware of large-scale natural datasets that provide ground-truth labels for all the desired spatial latents in a 3D coordinate frame. Therefore, we believe our approach using a synthetic dataset remains the most practical and effective way forward for this kind of study. We agree that comparing models trained on natural image datasets, such as the SA-1B segmentation dataset, is a compelling direction for future research. However, this falls outside the scope of our current work. Due to the limited time available for revisions, we were unable to include these results but have incorporated this suggestion into the discussion section as a potential avenue for future work. Additionally, our neural alignment analyses rely primarily on regression-based metrics, which are widely used within the community. While incorporating alternative metrics such as Representational Similarity Analysis (RSA) could provide additional insights, we believe it will unlikely alter our main findings. Given the time constraints of the revision process, we have deferred this to future research and added a discussion of this limitation and potential future directions in **line 523** of the new pdf.
>
> We hope our response answers your questions and addresses your concerns. If you have any other questions, please let us know!

---

### Official Review · Reviewer_v5sv · 2024-11-06

**Soundness:** 3
**Presentation:** 3
**Contribution:** 3
**Rating:** 6
**Confidence:** 4

**Summary:**

Traditionally, neural network based models of the visual cortex have relied on CNN backbones trained for object recognition. Here, the paper investigates if CNNs trained with alternate objectives (specifically, spatial tasks like viewpoint and pose estimation) can be used instead, and how that impacts the learned representations. Their main finding is that models trained to predict such alternate spatial tasks can also be used to learn models of the visual cortex which perform just as well. Digging into it further, the authors find this happens because that the representations learned with these different objectives all end up learning quite similar representations (as measured by CKA). Also, authors find that variability in non-target latents helps models.

**Strengths:**

1. **Very important problem**:
   The paper broadens our understanding of the ventral stream by exploring if spatial tasks, not just object categorization, can also drive effective visual cortex models. This is impactful for both building better models of the cortex, and also provides a computational feasibility argument for better understanding what the cortex does in the brain.

2. **In-depth, exhaustive analysis**:  The authors provide thorough testing across multiple objectives, using extensive synthetic data and well designed metrics. Largely, the evidence matches the claim, and the findings seem quite reliable.

3. **Interesting findings**:  Discovering that spatial-task-trained models achieve similar neural alignment to category-trained ones suggests surprising flexibility in modeling the visual cortex. It might opens an interesting question---how do self-supervised models of the visual cortex behave in comparison to categories and spatial latents?

4. **Interesting new dataset**:   The synthetic dataset with spatial labels could be of use to future researchers.

5. **Writing and figures are high quality**: Clear writing and well-designed figures make complex ideas accessible and enhance the paper’s readability.

**Weaknesses:**

1. Missing comparison to self-supervised models: The paper mentions them briefly, but does not provide experiments with these models. The best performing modern models used in AI are all increasingly relying on self supervised objectives.

2. Missing comparisons with Vision Language models: The majority of upcoming new models are all multi-modal. It would be interesting to see how similar or dissimilar these perform.

3. Missing some references:
- The ideas of non-latent diversity are closely related to work in machine learning focusing on Data Diversity. It would be nice to connect this work to these papers [1,2,3,4,5]
- This paper presents results on BrainScore which is in-distribution. It would be interesting to see how these evolves out of the training data distribution, especially since invariances become very important in out of distribution settings [6,7,8,9]

References

1. Koh, P.W., Sagawa, S., Marklund, H., Xie, S.M., Zhang, M., Balsubramani, A., Hu, W., Yasunaga, M., Phillips, R.L., Gao, I. and Lee, T., 2021, July. Wilds: A benchmark of in-the-wild distribution shifts. In International conference on machine learning (pp. 5637-5664). PMLR.

2. Madan, S., Henry, T., Dozier, J., Ho, H., Bhandari, N., Sasaki, T., Durand, F., Pfister, H. and Boix, X., 2022. When and how convolutional neural networks generalize to out-of-distribution category–viewpoint combinations. Nature Machine Intelligence, 4(2), pp.146-153.

3. Gulrajani, I. and Lopez-Paz, D., 2020. In search of lost domain generalization. arXiv preprint arXiv:2007.01434.

4. Arjovsky, M., Bottou, L., Gulrajani, I. and Lopez-Paz, D., 2019. Invariant risk minimization. arXiv preprint arXiv:1907.02893.

5. Arjovsky, M., 2020. Out of distribution generalization in machine learning (Doctoral dissertation, New York University).

6. Ren, Y. and Bashivan, P., 2024. How well do models of visual cortex generalize to out of distribution samples?. PLOS Computational Biology, 20(5), p.e1011145.

7. Madan, S., Xiao, W., Cao, M., Pfister, H., Livingstone, M. and Kreiman, G., 2024. Benchmarking Out-of-Distribution Generalization Capabilities of DNN-based Encoding Models for the Ventral Visual Cortex. arXiv preprint arXiv:2406.16935.

8. Pierzchlewicz, P., Willeke, K., Nix, A., Elumalai, P., Restivo, K., Shinn, T., Nealley, C., Rodriguez, G., Patel, S., Franke, K. and Tolias, A., 2024. Energy guided diffusion for generating neurally exciting images. Advances in Neural Information Processing Systems, 36.

9. Ponce, C.R., Xiao, W., Schade, P.F., Hartmann, T.S., Kreiman, G. and Livingstone, M.S., 2019. Evolving images for visual neurons using a deep generative network reveals coding principles and neuronal preferences. Cell, 177(4), pp.999-1009.

**Questions:**

1. Were you able to decode categories from the models trained for spatial latents?

---

> ### Author Response · Authors · 2024-11-12
>
> Thank you, reviewer v5sv, for your thoughtful reviews and for pointing us to the many interesting related works.
>
> **Response to your questions:**
> We were able to decode categories to some extent from the latent trained models. For example, in Figure 5b, we can decode the categories well above chance in a 117-way classification task. In Figure C.1, C.2, and Table C.1, C.2, we analyzed the behavior alignment scores of these latent trained models. This score measures how well the penultimate layer of the model supports categorization in a human-like way (Rajalingham et al., 2018). In summary, we found that spatial latent trained models are not as good as category-trained models in supporting categorization behavior if we decode from the penultimate layer. This may or may not hold in earlier layers; further analysis is needed.
>
> We are working on addressing your other comments, so stay tuned. Thank you!
>
> References:
> Rajalingham, Rishi, et al. "Large-scale, high-resolution comparison of the core visual object recognition behavior of humans, monkeys, and state-of-the-art deep artificial neural networks." Journal of Neuroscience 38.33 (2018): 7255-7269.

---

> ### Author Response · Authors · 2024-11-29
>
> We want to thank you for your effort in reviewing and helping us improve this paper.
>
> - Category decoding. First, we want to expand on our previous response to your question about decoding categories from spatial latent-trained models. In the revised pdf, we included a new supplementary **figure E.1**, in which we compared the category decoding performance from spatial latent-trained and category-trained models. We found that the category decoding performance of spatial latent-trained models is similar to category-trained models in early to middle layers, but they start to diverge in late layers. Spatial latent-trained models become much worse at representing categories than category-trained models in late layers. This is consistent with our findings in Figure 4, which shows that the representations of spatial latent-trained models are very similar to category-trained models in early and middle layers and diverge in late layers.
> - References about data diversity. We also want to thank you for pointing us to these related references. We discussed how our studies could connect to data diversity and domain generalization studies in **line 498** of the new pdf.
> - How do our models perform out of distribution? In the main text, we mainly focused on results from public benchmarks on Brain-Score. The V1 and V2 benchmarks already use images that are out of the distribution of our model training. The V4 and IT benchmarks are based on grey-scale images and have different objects and backgrounds than our training images in TDW, which is also out of distribution to some extent. However, V4 and IT benchmarks still have similarities to our training data in TDW since they all involve synthetic objects superimposed on random backgrounds. In the revision, we further evaluated our ResNet-18 models on other non-public benchmarks in Brain-Score, many of which include natural images that are substantially out of our models’ training distribution. We showed the results in supplementary **figure C.5**. We found that ResNet-18 models trained on spatial latents performed comparably to category-trained models in most V1, V2, and V4 benchmarks. However, among the five IT benchmarks, spatial latent-trained models significantly underperformed category-trained models in two benchmarks (”SanghaviMurty2020.IT-pls”, ”SanghaviJozwik2020.IT-pls”) while outperforming them in one (”Bracci2019.anteriorVTC-rdm”). These three IT benchmarks are based on natural images, which differ considerably from our synthetic training data. We believe our results based on public benchmarks provide a valuable foundation and invite future investigation into how models trained on different targets generalize to out-of-distribution images when evaluating neural alignment. We also discussed the limitations of our current results, including references to out-of-distribution generalization, in **line 535** of the new pdf.
> - Comparison to self-supervised models and vision language models. Thank you for your suggestion. We agree that comparing our models to self-supervised and vision-language models is an intriguing and valuable direction for further exploration. However, we believe that conducting such experiments, while insightful, lies outside the scope of the core claims and focus of our current work. Our primary focus has been investigating category-trained and spatial latent-trained models, drawing from the “what” and “where” frameworks discussed in cognitive science and neuroscience. Due to time constraints during the revision process, we were unable to perform a thorough and meaningful comparison with these models. We view this as an excellent avenue for future research and hope that our findings will inspire further investigations in this direction.
>
> We hope our response answers your questions and addresses your concerns. If you have any other questions, please let us know!

---

### Comment · Area_Chair_fus8 · 2024-11-26
**[IMMEDIATE ACTION NEEDED] Respond to author rebuttal**

Dear Reviewers,

We are reaching the end of the discussion period (November 26 at 11:59pm AoE): please check the author rebuttal and post your response at your earliest convenience. Please also update your review accordingly, even if your view of the paper has not changed (to acknowledge that you have read the rebuttal).

Thank you,
--Your AC

---

### Meta-Review · Area_Chair_fus8 · 2024-12-20

**Metareview:**

This paper studies alignment of visual representations between artificial convolutional neural networks (CNNs) and primate visual areas (from neural array recording data of primates observing visual stimuli). The authors study whether CNNs trained with tasks that involve prediction of spatial variables (e.g. object pose) on solely synthetically generated data can give rise to high alignment scores. The paper finds that models trained this way (using just a few predicted latents) achieve comparable alignment to models trained on multi-category image classification (with hundreds of categories).

The reviewers all agree that this is an interesting paper, it studies an important problem, is well-executed, and arrives at an interesting, relevant and novel conclusion. Concerns around experimental evaluation, baselines, and methodology were addressed in the rebuttal.

This paper will make a great addition to the conference. The authors are encouraged to take the reviewer feedback into account when preparing the camera-ready version of the paper.

**Additional Comments On Reviewer Discussion:**

Reviewer consensus did not change during discussion.

---

### Decision · Program_Chairs · 2025-01-22

Accept (Poster)